

# Use of a remotely piloted aircraft system for hazard assessment in a rocky mining area (Lucca, Italy)

Riccardo Salvini[1], Giovanni Mastrorocco[1], Giuseppe Esposito[1], Silvia Di Bartolo[1], John Coggan[2],
Claudio Vanneschi[2]

[1]Department of Environmental, Earth and Physical Sciences and Centre of GeoTechnologies, University of Siena, Via Vetri
Vecchi 34, 52027 San Giovanni Valdarno, AR, Italy
[2]University of Exeter, Camborne School of Mines (CSM), College of Engineering, Mathematics and Physical Sciences
(CEMPS), Penryn, Cornwall TR10 9EZ, UK

*Correspondence to*: Riccardo Salvini (riccardo.salvini@unisi.it)

**Abstract.** The use of remote sensing techniques is now common practice in different working environments, including engineering geology. Moreover, in recent years the development of structure from motion (SfM) methods, together with rapid technological improvement, has allowed the widespread use of cost effective remotely piloted aircraft systems (RPAS) for acquiring detailed and accurate geometrical information even in evolving environments, such as mining contexts. Indeed, the acquisition of remotely sensed data from hazardous areas provides accurate 3D models and high resolution orthophotos minimizing the risk for operators. The quality and quantity of the data obtainable from RPAS surveys can then be used for inspection of mining areas, audit of mining design, rock mass characterizations, stability analysis investigations and monitoring activities. Despite the widespread use of RPAS, its potential and limitations have still to be fully understood.

In this paper a case study is shown where a RPAS was used for the engineering geological investigation of a closed marble mine area in Italy: direct ground based techniques couldn't be applied for safety reasons. In view of re-activation of the mining operations, high resolution images taken from different positions and heights were acquired and processed by using SfM techniques, for obtaining an accurate and detailed three-dimensional model of the area. The geometrical and radiometrical information was subsequently used for a deterministic rock mass characterization that led to the identification of two large marble blocks that pose a potential significant hazard issue for the future workforce. A preliminary stability analysis was then carried out in order to demonstrate the potential use of RPAS information in engineering geological contexts for geo-hazard identification, awareness and reduction.

## 1 Introduction

In open-pit areas, personnel and equipment involved in mining operations can be exposed to different types of slope instability processes. Rock collapses can be due to a series of predisposing and triggering factors, mostly depending on relationships between localized geological conditions and mining activities. According to Zajc et al. (2014), for example, hazardous situations may occur when sedimentological characteristics and geological structures (e.g. joints, faults, bedding



planes) of rock masses are altered by exploitation. At the same time, Zheng et al. (2015) underlain the crucial role played by morphological features, like sharp cuts and steep slopes, for the triggering of rockfalls in mining areas. As widely demonstrated in the literature, the understanding of geometric relationships between geological discontinuities and slope morphology is essential to evaluate the potential occurrence of rock failures, since orientation of joint sets may influence

both the size and failure mechanisms of rock blocks prone to collapse (e.g. Stead and Wolter, 2015). Generally, discontinuity characterization is carried out in the field by traditional engineering geological surveys (Priest, 1993); measurements may be subjected to different source of errors which can result in under- or over-estimation of the discontinuity geometrical properties (Tuckey and Stead, 2016). In order to avoid this deficiency Sturzenegger and Stead (2009) suggested to couple traditional field measurements with remote sensing techniques. Indeed, techniques such as terrestrial laser scanning (TLS)

and digital photogrammetry (DP) for rock mass characterization are increasingly being used, especially in open pit mines where rock slopes subjected to excavation are analyzed (e.g. Kovanič and Blišťan, 2014; Salvini et al., 2015; Tuckey and Stead, 2016). TLS and DP allow accurate representation of rocky outcrops by means of 3D point clouds or interpolated models. However, it is worth noting that ground-based acquisition of high resolution topographic data of complex morphologies may be very difficult to acquire because of occlusions and inaccessible zones (Passalacqua et al., 2015). A

solution to this problem is provided by the use of remotely piloted aircraft systems (RPAS) as a platform to acquire light detection and ranging (LiDAR) or photogrammetric data. According to Chen et al. (2015) there are only few published references related to RPAS applications in open-pit mining. The majority of photogrammetric studies deal with geomorphic feature characterization or mapping of the surface mine extent (Lamb, 2000; Chen et al., 2015; Shahbazi et al., 2015; Tong et al., 2015; Esposito et al., 2017). Few works concern the use of RPAS for discontinuity characterization of rock slopes

affected by mining activity. Salvini et al. (2016), for example, used an RPAS to map discontinuities in a marble quarry and to subsequently build 3D discrete fracture network models. McLeod et al. (2013) explored the feasibility of using RPAS-acquired video images to derive 3D point clouds and to measure fracture orientations. Digital images acquired from RPAS are commonly processed with the structure from motion (SfM) technique (Spetsakis and Aloimonos, 1991; Fonstad et al. 2013; Colomina and Molina, 2014; Westoby et al., 2012). SfM is based on sophisticated algorithms of image matching that

use pseudo-random redundant images acquired from multiple viewpoints to reconstruct the three-dimensional geometry of an object or surface. In order to analyze rock outcrops, the use of RPAS multicopters results particularly suitable because they allow different geometric configurations for the image acquisition (i.e. zenithal, frontal, oblique). Multiple images obtained from different angles help both the image alignment procedure and limit non-linear deformations. Moreover, the relatively short distance to which multicopters can operate from rock faces allows acquisition of high resolution images that

can be used for producing high quality topographic products. However, in RPAS-SfM applications particular care is needed when geo-referencing the 3D model. As stated by Passalacqua et al. (2015), sensors such as cameras or lasers fixed to RPAS typically do not have onboard navigation systems with a sufficient accuracy for geodetic positioning. In fact, the global navigation satellite system (GNSS) and inertial measurement unit (IMU) devices typically mounted on RPAS are used for navigation and flight stabilization purposes and allow only a rough airborne cameras exterior orientation (Gonçalves and



Henriques, 2015). In order to obtain accurate 3D models, the use of ground control points (GCPs) surveyed with geodetic GNSS receivers and total station (TS) is generally employed (Francioni et al. 2015). Nevertheless, the final accuracy is dependent not only from the GCP-related accuracy, density and distribution within the surveyed area, but also from image quality and percentage of overlapping between single frames. Therefore, careful planning of an RPAS photogrammetric

survey plays a crucial role in providing accurate results necessary for subsequent analysis, such as determination of discontinuity measurements.

In this study, two RPAS-based photogrammetric surveys were carried out within an open-pit mine of the Apuan Alps marble district, Italy. These surveys aimed to obtain detailed topographic information of the area. The 3D data was then used to perform a preliminary hazard evaluation, requested in view of a potential restart of the mining operations interrupted some

years ago. Indeed, the safety of the workforce represents a critical aspect for the exploitation of the marble quarries of the Apuan Alps. In the last decades, many deadly rock failures involving personnel employed in the mining activity have occurred. The last accident occurred on April 14, 2016 (Petley, 2016). In this case, two workers were killed and another injured by a large rockfall involving around 2000 tons of marble, during the excavation of a fractured rock wall. The geo-structural conditions of the marble predispose the rock masses to different types of failures with different magnitudes. Slope

stability analysis are therefore essential to improve safety conditions for personnel employed in the mines. However, a complete analysis of all the slopes characterizing an open-pit mine is often problematic, given their spatial extension and limitations of numerical models. For this reason both geological and geomorphological information of the whole mining area are essential to detect and evaluate the most hazardous situations. RPAS-derived data were therefore integrated with those acquired in the field from a traditional engineering geological survey. The combined use of these information allowed

preliminary 3D analysis and evaluation of the stability conditions of a large rocky block that posed a risk to the mining area.

## 2 Geographical and geological setting

The study area is located in the Apuan Alps marble district, in the province of Lucca (Tuscany, Italy), Fig. 1. The open pit, named "Piastrone", is characterized by a V shape, with two principal slope directions oriented 50/90 and 323/90 (dip direction/dip). The bottom of the pit is located at 1,180 meters a.s.l., but the excavated rock faces can reach and overcome

1,300 meters a.s.l.. The rock mass is characterized by different sets of discontinuities with persistence values vary from few meters up to decameters.

From a geological point of view the Piastrone open pit is located in the Apuan Alps metamorphic complex, precisely in the Mt. Altissimo Syncline (AS), belonging to the Apuane Unit (Meccheri et al., 2007), Fig. 2. According to classical interpretation (Carmignani and Kligfield, 1990) AS resulted from a compressive tectonic phase originated during the

Tertiary continental collision between the Sardinia-Corsica block and the Adria plate. Successively, during the Early Miocene, a new ductile to brittle-ductile deformation caused by a post-compression tectonic uplift overprinted the earlier structures and generated a widespread network of joints and faults. In the Mt. Altissimo area, the main set of fragile



deformation strikes SW-NE to W-E with sub-vertical dip, generally with negligible motion except from few cases where offsets of some ten meters have been observed (Meccheri et al., 2007).

AS involves the oldest terms of the Apuane unit sequence, with pre-Alpine basement rocks, Grezzoni dolostones, megalodont-bearing marbles with metabreccias and chloritoid-rich phyllites, local lenses of dolomitic marblems and Marbles

sensu stricto of lower Liassic age (Meccheri et al., 2005). Due to the compressive tectonic phase, a penetrative S1 foliation is also present in all the lithotypes (except from dolostones).

# 3 Methods

## 3.1 Geomatic survey

In order to assess and localize the slope stability hazard in the rocky mining area, two RPAS surveys were carried out in

zenithal modality and in a parallel direction to the rock faces (frontal). The surveys were performed in December 2015 using the Aibotix$^{TM}$ Aibot X6 V1 multicopter, composed by six electric rotors, and equipped with a Nikon$^{TM}$ CoolpixA digital camera (Table 1) and a GNSS/IMU system that allows recording of 3D coordinates (X0,Y0,Z0) and orientation of the camera (pitch, roll and yaw – ω ϕ κ) at every shoot.

The zenithal survey was preliminarily designed in laboratory with the Aibotix$^{TM}$ Aiproflight planning software, and

manually performed through single quasi-parallel flight lines. A total of 151 aerial images were acquired with a nominal overlap and sidelap of 80% and 60% respectively. Two flights were needed to cover all sectors of the mining area (Fig. 3).

An average estimated ground sampling distance (GSD) of 2.4 cm was calculated. During the flight, a GNSS field survey was performed in order to ensure the necessary spatial accuracy for the images exterior orientation, measuring a total of 8 artificial targets 50x50 cm large uniformly distributed over the study area (Fig. 4) and used as ground control points (GCPs)

and check points.

The GNSS survey was carried out in Real Time Kinematic (RTK), using geodetic receivers. In particular, a reference station was set up, recording continuous signals from the GNSS satellite constellation above 3 hours. The positional information acquired by the reference station was then sent to a mobile receiver, using a radio modem communication. Each GCP was occupied for at least two minutes with a recording interval equal to 1 sec. The coordinates of the points acquired with this

technique were corrected by post-processing procedures using contemporary data recorded by three permanent GNSS stations (La Spezia, Pieve Fosciana and Pisa) allowing centimetric accuracy. The orthometric heights were also calculated by using Convergo, an Italian code for full coordinates conversion. The coordinates of the GCPs were collected in ETRF2000 and then converted in the Italian National Gauss Boaga system for the exterior orientation of the images and the restitution of slopes and joints orientation.

The frontal survey, with directions of acquisition parallel to the rock faces, was carried out manually, without the use of the Aibotix$^{TM}$ AiProflight planning software. Six flights were needed to cover all sectors of the mining area, for a total of 448





overlapping images. The flights were executed according to sub-parallel straight lines about 60 meters distant from the rock face (Fig. 5) providing an average estimated GSD of 1.5 cm.

As for the zenithal flight, a series of GCPs and check points (21 targets in total - Fig. 6) were measured by a reflectorless TS. Due to the complex morphology of the slopes and the extent of the mining area, an high number of GCPs was used in the

photoprammetric process. Two GNSS receivers (working in static mode) were used to collect geographic coordinates of two points necessary for the roto-translation of the measured GCPs: TS survey origin and zero-Azimuth direction. Also for this survey, GNSS data were post-processed using contemporary data acquired by permanent GNSS stations and ellipsoidal heights were converted to orthometric heights.

## 3.2 Application of structure from motion algorithms

The software Agisoft[TM] PhotoScan Professional version 1.2.5 (Agisoft 2016) was used to process the images obtained with the two RPAS surveys (two zenithal flights plus six frontal flights). This software is able to resolve the camera equations (interior and exterior orientation) and to generate georeferenced spatial data like three-dimensional point clouds, digital surface models (DSMs) and orthophotos. All the images acquired in the two surveys were processed with an identical photogrammetric processing, in two distinct Agisoft[TM] Photoscan projects, one for the zenithal flights and another for the

frontal flights .

The first step consisted in the image alignment, through which the interior and relative orientation parameters were solved. In order to improve the whole alignment process and to obtain low reprojection error, millions of tie points were automatically extracted without setting a point limit. As result of the previous stage, all images were aligned. The second step involved georeferencing of the 3D model in such a way as to solve the exterior orientation parameters by using the

GCPs coordinates measured during the two GNSS-TS topographic surveys. For both surveys, a part of measured points was used as check points to verify the model accuracy. Specifically, for the zenithal survey 2 of the 8 measured points were used as check points, while for the survey with directions of photo acquisition parallel to the rock faces, 4 points out of 21 were used as check points. Both natural and artificial targets were identified directly on the images, assigning a 3D coordinate to each of them.

Subsequently, the optimize tool was utilized in order to remove possible non-linear deformations, reducing re-projection errors and possible photos misalignment. Moreover, the optimization was improved by deleting all the tie points with a reprojection error greater than 1 pixel.

In a subsequent step, the zenithal and frontal dense 3D point clouds were generated with medium quality and aggressive depth filtering settings. No automatic classification of clouds was necessary: no infrastructure was present as the mine was

not operational and there was no vegetation within the area of interest.

Lastly, a polygonal 3D mesh model was created from the point cloud and used to create the orthophoto of the open pit area. The orthophoto has the property of being georeferenced, 'scale-corrected' and removed from image distortions, therefore true



distances can be measured. The corrected image with a spatial resolution of 1 cm/pixel was finally projected into the Italian National Gauss Boaga system.

### 3.3 Engineering geological investigation

In order to characterize the open pit mine, a relatively large number of discontinuities was identified directly on the dense point cloud. The orientations of the selected joints were manually determined by using patches that reproduce every single discontinuity plane in the point cloud using the LeicaTM Cyclone 9.0 software. The discontinuity sets were then recognized using stereographic representation (Schmidt equal-area method, lower hemisphere).

According to Mastrorocco et al. (2017), a manual deterministic fracture mapping was adopted because it increases the level of control of the process, that is essential where the morphology of the quarry surfaces is largely artificial (smooth cut surfaces). The collected data were then compared with those manually measured by traditional engineering geological surveys. On the basis of engineering geological data, the geological strength index (GSI - Hoek and Brown, 1997) and the rock mass rating (RMR - Bieniawski, 1989) were applied, and a kinematic stability analysis was carried out using the Markland test (Markland, 1972). The latter testing was executed in order to identify potential kinematic mechanisms that characterize the slopes. The tests for planar sliding, wedge sliding, and direct toppling were performed for both principal slope directions (Eastern slope - dip direction/dip 50/90; Western slope - dip direction/dip 323/90).

Despite the importance of performing kinematic analysis to discover the possible block instability, one of the principal limitations of this stereographic method is the inability to locate the block source areas. For this reason, the most critical blocks have been identified directly on the point cloud. The points representing the geometry of every single block were meshed in Leica[TM] Cyclone 9.0 and their volume estimated in respect of reference planes corresponding to the discontinuities that demarcate or shape the respective blocks.

The collected data were finally used for preliminary stability analyses using Rocscience[TM] Swedge software. Swedge is a 3D software for evaluating the stability of surface wedges in rock slopes. It considers the intersection of discontinuities and allows the calculation of safety factors of formed blocks. The software is based on classical limit equilibrium methods that usually have some limitations, such as lack of consideration of in-situ stress, strains and intact material failure (Stead et al., 2006). Nevertheless, Swedge does allow the consideration of external forces, by applying a force (vector with given orientation and intensity) to the formed blocks. From this, preliminary analysis of the impact of water content or other forces can be performed.

## 4 Results

### 4.1 Photogrammetric modelling

The image alignment process, described in paragraph 3.2, resulted in a reprojection error related to the manual placement of GCPs on the images of 0.41 pixel for the zenithal survey and 0.48 pixel for the frontal survey. The final Root Mean Square





Error (RMSE) for the zenithal flights exterior orientation was equal to 0.042 meters; RMSE for the frontal flights exterior orientation was equal to 0.043 meters (Table 2).

The final 3D frontal and zenithal point clouds are constituted by more than 18,000,000 and almost 13,000,000 of points respectively, with a mean point spacing varying from 1 to 4 cm.

## 4.2 Rocky slope engineering geological characterization

The orientation of 154 discontinuity planes manually selected on the point cloud was calculated through stereographic projection. The final stereonet allowed to identify four discontinuity sets, whose properties are listed in table 3. Figure 7 shows a comparison between the latter (b) and the stereonet obtained through traditional manual engineering geological survey (a), highlighting a good level of congruence that confirms the quality of the taken approach.

Based on the discontinuity characteristics derived from the RPAS surveys, the basic RMR (RMRb) and GSI index were calculated. The RMRb was found to be 67, while the GSI was estimated to be between 60 and 65, indicating in both cases a rock mass of good quality. These results agree with the authors' field observations and what is described in the actual quarry excavation plan (Lorenzoni, 2012). In view of the rock competency, potential instability is not related to general weaknesses of the rock mass, but the intersection of discontinuity planes can locally isolate rocky blocks with the potential for sliding or toppling. For this reason, a kinematic stability analysis was performed. A discontinuity friction angle of 35° was considered: this agrees with data from quarry's advisors studies (Lorenzoni, 2012; Dumas, 1999), by the Geomechanical laboratory of the Centre of GeoTechnologies of Siena University, and literature (Chang et al., 1996, Perazzelli et al., 2009, Mastrorocco, 2013) and. Table 4 shows the potential unstable discontinuity sets obtained from kinematic stability analysis (examples are shown in Fig. 8) for both principal slope orientations.

Results highlight the potential for blocks of variable shape and size, varying from about a cubic meter to a few hundred cubic meters, that may be subject to gravity induced instability. Therefore, the high resolution images and the dense point cloud were analyzed in order to locate possible block source areas. More than 20 blocks were deterministically characterized in terms of size, shape and barycentric coordinates. The adopted approach also identified two large blocks, a few thousand cubic meters in size, with potential for sliding. These are shown in Fig. 9, and are formed by the intersection of two different faults and a discontinuity basal plane with 5 cm aperture, no infill, smooth surface and high persistence. The first block, Block A, is of particular interest because it daylights in the face and prevents Block B from sliding (similar to an active-passive wedge; Prandtl's prism transition zone - Kvapil and Clews, 1979). A main road access to the quarry is located at the base of this slope, increasing the potential risk for the area.

The geometric characteristic of the two blocks, including orientation of the intersecting discontinuities and volume of the meshed block, were obtained using Leica™ Cyclone 9.0 and are shown in table 5.

Due to the particular geometrical configuration, Block A can be described as the key block as it daylights the rock face. In the actual setting Block B does not hold the potential for sliding as it does not daylight in the slope face, but it could play a significant role in terms of additional weight force. Nevertheless, the following preliminary stability analysis is focused on





Block A. Further investigation would require an analysis of the effect of Block B on the potential for instability as this provides the 'active' component of the active-passive wedge.

**4.3 Preliminary slope stability analysis**

The geometry of Block A was deterministically re-created in Swedge using the geometrical information obtained from the point cloud provided in table 5. Initially the discontinuities were assumed to be fully persistent. This is a common approach in engineering geology, since reliable values of persistence are almost impossible to obtain from field mapping and most rock slope stability analysis assume that the 100 % persistent joint exists on failure surface (Park, 2005). Moreover, in this case two discontinuities (lateral and back surfaces) correspond to geological faults and can be therefore considered fully persistent. However, the basal plane is a joint and the possible presence of rock bridges should be carefully considered. In general, joint persistence (K) is defined as the fraction area that is actually discontinuous (Einstein et al., 1983), and can be calculated with the following Eq. (1):

$$K = \lim_{A_D \to \infty} \frac{\sum_i a_{Di}}{A_D} \tag{1}$$

where D is a region of the plane with area $A_D$ and $a_{Di}$ is the area of the joint in D.

The limit of the application of this method is that the discontinuity area is practically impossible to be determined in the field. For this reason, Einstein et al. (1983) proposed a rough quantification of persistence value by measuring trace length on a rock exposure. Jennings (1970) proposed the following Eq. (2) for persistence calculation starting from trace length values on rock exposure:

$$K = \frac{\sum JL}{\sum JL + \sum RBR} \tag{2}$$

where JL is the total length of the joints segment and RBR is the total length of rock bridges.

In this case the basal plane did not show the presence of segments of intact rock along its trace on the rock exposure, consequently Eq. (2) confirms a 100 % persistence of the basal plane, that was used in the first analysis.

The adopted limit equilibrium solution for the slope stability analysis was based on the Mohr-Coulomb shear strength model with a friction angle of 35° and a unit weight of 0.026 MN/m³ (Ertag, 1980, Dumas, 1999). It should be noted that the western lateral surface observable in the model was also necessary to re-create the block geometry in the software. It was assigned 0° friction angle so as not to induce a resisting force in the simulation. Water forces were also initially ignored within the preliminary analysis. The result of the analysis is shown in Fig. 10.

The result highlights a possible condition of instability for Block A which does not match with field observations, since the block under study has remained stable in this position for tens of years. In order to investigate the effect of uncertainty or variability of the input parameters, a sensitivity analysis was performed. In sensitivity analysis specific parameters are varied across a range of values and the effect on safety observed factor. This helps to identify the parameters that have the most effect on block stability. Since the geometrical inputs are well defined through the accurate 3D model, the subsequent analysis focused on waviness angle (it accounts for the undulations of the joint surface, observed over distances on the order



of 1 m to 10 m; Miller, 1988), cohesion and friction angle of the basal plane and water pressure. These are also the parameters with the higher input uncertainty. Figure 11 shows the result of the sensitivity analysis performed on the cited values.

As observed, the cohesion is clearly the parameter that has the most effect on block stability. For this reason, the effect of

this parameter was investigated in more detail in the following analyses. In practice, in rock slopes, the cohesion of intact rock bridges between discontinuous joints increases the shear strength of the surface. This can be one to two orders of magnitude greater than the shear strength available on the discontinuity (Park, 2005). Mathematically, it is possible to consider the presence of rock bridges in terms of effective cohesion along the shear surface (Eberhardt et al., 2004) by using the following Eq. (3):

$$c_i = c \frac{A_g}{A}$$     (3)

where c is the intact rock cohesion, $A_g$ the total area of intact rock bridges along the shear surface, and A is the total area of the shear surface. Application of Eq. (3) makes it possible to determine an effective cohesion dependent on the continuity of jointing. From this the contribution of eventual rock bridges on the block stability can be investigated starting from intact rock cohesion material value, that has been determined to be approximately 16 MPa (Lorenzoni, 2012; Dumas, 1999; data

from Geomechanical laboratory of the Centre of GeoTechnologies of Siena University). Table 6 shows the results in terms of factor of safety obtained from a parametric instability analysis performed with increased values of effective cohesion, corresponding to 0, 0.5, 1, 2, 5, and 10 % of rock bridges on the basal plane (total area of 510 m$^2$), and a 20 % of water filled fissures (considered reliable after field observation and high resolution images analysis).

**5 Discussion**

The RPAS approach adopted in this case study, based on the acquisition of high-resolution images from different perspective and accurate GNSS/TS topographic surveys, overcame problems related to image acquisitions in presence of steep and high quarry walls and provided high-resolution orthophotos of the site (1 cm pixel size). The application of RPAS instrumentation was extremely successful for the reconstruction of the complex morphology of the mine site where ground based techniques (e.g. terrestrial laser scanning, terrestrial photogrammetry) have limitations due to potential "shadow" effects and several

inaccessible set-up zones due to safety reasons. GCPs measured by the combined use of TS and GNSS receivers permitted to reach an optimal external orientation of the images, fundamental for any accurate discontinuity measurement. On the other hand, the use of RPAS can be limited by the need of a pilot license and operator experience on topographic survey and SfM processes. Indeed, the accuracy of the final 3D model may be greatly influenced by the quality of data collected, hardware and software capacity and user experience.Although several authors successfully proved the applicability of automatic and

semi-automatic algorithms for fracture mapping from images and three-dimensional point clouds (Mah et al. 2011; Vöge et al. 2013; Assali et al. 2014; Vasuki et al. 2014), a complete manual approach was adopted in this analysis because in most cases the flat morphology of the mine rock faces only permits photointerpretation of discontinuity traces. This approach



guarantees high quality and reliability of the results, also considering that a visual inspection of the outputs in this contexts is always suggested, even when using automatic approaches (Salvini et al., 2016). Therefore, the orientation of several discontinuity planes was calculated using Leica$^{TM}$ Cyclone 9.0 on the point cloud, once its high positional accuracy level was demonstrated. This allowed for a more complete characterization of the rock mass than the one that can be obtained through

traditional engineering geological survey, due to limited safe access to the slopes within the site. This was particularly important also because discontinuities characteristics can be different at diverse heights of the rock mass because of stress relaxation caused by mining activity. In this context, the possibility to inspect the mining area from different angles in high definition, allowed identification of critical areas to be analyzed in detail for safety purposes. Moreover, the possibility to use the point cloud for obtaining geometrical characteristics of blocks represented a major advantage, because it allowed the

exact geometrical reconstruction of a 3D model to be used in specific software for slope instability analysis. In this work a potential significant risk was identified for the future workforce due to the presence of two major blocks with potential for sliding. In fact, with conservative assumptions the preliminary limit equilibrium analysis showed that the key Block A, in its present shape, is potentially unstable. This is mainly due to the fact that the basal plane dips out of the slope and daylights on the face, with a dip angle higher than the friction angle of the surface. Moreover, the block is separated from the rock mass

by a major fault, that can be simulated in the Swedge analysis as a tension crack. The fault can be clearly identified from the orthophoto obtained from the application of SfM method. Figure 12 shows an apparent motion identified on the back fault. From a geological point of view the fault can be contextualized in a East-West system that characterize this area of the Apuan Alps complex, as observable in Fig. 2. The presence of cataclasite with variable thickness can be interpreted as another sign of movement that released the block from the rock mass, similarly to the fault that acts as lateral release

surfaces.

In this context the major uncertainty is on the basal plane (Fig. 12). Despite the presence of a continuous trace line on the rock exposure, its full persistence in the rock mass is not clear. In general, the presence of rock bridges plays an important role in stabilizing the removable rock blocks. In particular, a rock block cannot fall or slide from a slope until the rock bridges have failed. The rock bridge failure involves the collapse of the intact rock, which can be an order of magnitude

stronger than the rock mass (Kemeny, 2005). From the sensitivity analysis, cohesion of the basal plane was the parameter that has the most significant influence on the block factor of safety. This suggests that it may not be completely persistent in this case, since the block has remained stable over time. In this regard, the parametric analysis carried out increasing the cohesion values shows how a small rock bridge, corresponding to 1% of the basal plane surface (5.1 m$^2$ of intact rock) is sufficient for guaranteeing the stability of the block. This is mainly due to the fact that, despite the basal plane dips more

than the friction angle of the surface, its inclination is not sufficient to avoid the generation of a high normal force that increases the shear strength of the discontinuity. Therefore, even a small area of intact rock increases the resisting force that leads to a condition of stability. In this case, 5.1 m$^2$ of rock bridge corresponds theoretically to a resisting force of 81.7 MN. In reality, based on field observation and authors' experience in similar contexts, higher percentage of rock bridges may exist, that could lead to increased safety. Nevertheless, Hudson and Priest (1983) identified two kinds of persistence relative





to impersistent or intermittent joints that should be considered. Differently from impersistent joints, intermittent discontinuities require a network of joint segments and intact regions on the same plane. However, as described by Mauldon (1994), the formation of intermittent joints is geologically unlikely, unless weakness planes exist within the rock mass. From this it follows that the cohesion of rock bridges in intermittent joints could be much lower than that of the intact rock. This

could be the case of Block A, and the presence of a series of discontinuities with similar dip and dip direction to the basal plane, observable in Fig. 13, seems to confirm the hypothesis of a preferential plane of weakness due to the geomechanical characteristics of the marble material in that portion of the mining area.

Moreover, the progressive degradation with time of rock bridges elements could cause a progressive failure mechanism that has the potential to lead to a final rockfall event. This is particularly important in small engineered slopes such as the present

one, where the rock mass may be continuously disturbed by excavation activity driving the slope to instability. Such mechanisms of progressive brittle fracturing of rock bridges are not considered in limit equilibrium approaches, and it is a known key limitation (Tuckey and Stead, 2016). The result is that using the Mohr-Coulomb shear criterion, inclusion of a small content of rock bridges adds significant apparent cohesion to the failure surface (Elmo et al., 2011; Tuckey and Stead, 2016).

The aspects discussed in this section lead to the conclusion that a potentially hazardous situation should not be underestimated. Therefore, in case of re-opening of mining activities an in-depth engineering geological analysis, together with the installation of a monitoring system for observing the behavior of the rock mass over time should be considered.

## 6 Conclusion

The case study highlights the powerful use of RPAS technology for rock slope characterization and acquisition of accurate

3D data for subsequent instability analysis. Specifically, an Aibotix$^{TM}$ Aibot X6 six-rotor multicopter was employed to obtain high resolution topographic data of a blocky rock mass located within a quarry prone to discontinuity-controlled instability mechanisms. A detailed 3D model of the area allowed accurate identification and geometrical measurement of the geological discontinuities that isolate significant volumes of rocks. The stability analysis performed with Rocscience$^{TM}$ Swedge software showed that rock bridges have a significant influence on stability conditions. The analysis highlighted the

need for further detailed analysis and installation of suitable monitoring systems for future quarry operations.

These results confirm the reliability of the employed technologies to provide data for preliminary evaluation of the hazard affecting the study area. The RPAS allowed acquisition of high resolution topographic data in an area characterized by a complex morphology where ground based techniques would have significant limitations (e.g. terrestrial laser scanning, photogrammetry). It is worth noting that in mountainous environments, the use of RPAS has to be evaluated according to the

local atmospheric and topographic conditions. The high temporal and spatial variability of the atmospheric conditions at high altitudes, as well as the presence of vegetation or steep and irregular slopes, could endanger the flight operations. This requires pilots with relevant experience and RPAS equipped with innovative systems to manage emergency conditions.





Future analysis at this site will concentrate on the evaluation of the most useful countermeasures to reduce the risk conditions, by monitoring the unstable slopes and undertaking further instability analysis including more complex 3D discrete fracture network (DFN) evaluation to assess the effects of rock bridges and elasto-plastic numerical approaches to assess likely instability.

5  **Acknowledgments**

The present study was undertaken within the framework of an agreement with USL1 of Massa and Carrara (Mining Engineering Operative Unit - Department of Prevention). The authors acknowledge M. Pellegri and D. Gullì (USL1) and V. Lorenzoni (Professional geologist) for their support of this research.





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



**Table 1.** Utilized RPAS and camera specifications.

| RPAS Type | Dimensions (cm) | Engines | Rotor diameter (cm) | Empty weight (kg) | Max. takeoff weight (kg) |
|---|---|---|---|---|---|
| Aibotix Aibot X6 v1 | Width 105 Height 45 | Brushless motors | 30.48 (12") | 2.45 | 6.5 |
| **Camera** | **Sensor type** | **Sensor Size (mm)** | **Image size (pixel)** | **Pixel size (mm)** | **Focal length (mm)** |
| Nikon CoolpixA | CMOS | 23.6 x 15.6 | 4,928 x 3,264 | 0,0048 | 18.5 |



**Table 2.** Information related to the zenithal and frontal photogrammetric surveys and processing.

|  | ZENITHAL RPAS SURVEY | FRONTAL RPAS SURVEY |
|---|---|---|
| **Number of images** | 151 | 448 |
| **GSD[1]** | 0.024 m/pixel | 0.015 m/pixel |
| **Relative flying altitude** | 93.9 m | 60.7 m |
| **# Tie Point** | 1,484,605 | 3,783,992 |
| **GCP[2] RMSE[3]** | 0.042 m | 0.043 m |
| **Check Point RMSE** | 0.065 m | 0.03 m |
| **GCP reprojection error** | 0.41 pixel | 0.48 pixel |

[1]Ground Sampling Distance; [2]Ground Control Point; [3]Root Mean Square Error





**Table 3.** Characteristics of the discontinuity sets measured on the study area.

| Set | Dip Dir/Dip (degrees) | Aperture (mm) | Filling | Persistence (m) | Spacing (m) | JCS (MPa) | JRC | Weathering | Water |
|-----|----------------------|---------------|---------|-----------------|-------------|-----------|-----|------------|-------|
| K1 | 231/60 | 0-1 | None, hard filling | 2-10 | 0.1-0.3 | 50 | 2-6 | Slightly weathered | Damp |
| K2a | 234/86 | 0-0.5 | Hard filling | 5.5 | 5-10 | 60 | 2-4 | Slightly weathered | Dry |
| K2b | 66/86 | 0-0.5 | Hard filling | 5.5 | 5-10 | 60 | 2-4 | Slightly weathered | Dry |
| K3a | 142/81 | 0-2 | None | <20 | 10-15 | 50 | 2-6 | Slightly weathered | Damp |
| K3b | 177/84 | 0-2 | None | <20 | 10-15 | 50 | 2-6 | Slightly weathered | Damp |
| K4 | 291/67 | 1-2 | None | 1-3 | 0.5-1.5 | 55 | 2-4 | Slightly weathered | Damp |



**Table 4.** Potentially unstable discontinuity systems along the two different slope orientations.

| Slope | Planar sliding | Wedge sliding | Direct Toppling |
|---|---|---|---|
| **50/90** | K2b | K3a/K3b, K2b/K3a, K2b/K3b, K2b/K4, K2a/K4 | K3a/K4, K1/K3a, K1/K4 (basal plane K2b) |
| **323/90** | K4 | K1/K3b, K3b/K4, K1/K4, K2a/K4, K2b/K4 | K3a/K3b, K2b/K3a, K2b/K3b, K1/K2b, K1/K2a, K2a/K2b (basal plane K4) |





**Table 5.** Characteristic of identified blocks A and B.

| ID | Volume (m3) | Height (m) | Width (m) | Length (m) | Basal plane (Dip Dir/Dip) | Lateral release surface (Dip Dir/Dip) | Back discontinuity (Dip Dir/Dip) |
|----|-------------|------------|-----------|------------|---------------------------|---------------------------------------|----------------------------------|
| A | 2650 | 35 | 15 | 40 | 031/42 | 307/88 | 350/81 |
| B | 7500 | 40 | 20 | 32 | 031/35 | 307/88 | - |





**Table 6.** Results of the parametric analysis increasing basal plane cohesion values.

| Rock bridge % | Intact rock (m$^2$) | Total cohesion (MN) | Driving force (MN) | Resisting force (MN) | Factor of safety |
|---|---|---|---|---|---|
| 0 | 0 | 0 | 48.4 | 35.9 | 0.7 |
| 0.5 | 2.5 | 40.8 | 48.4 | 76.7 | 1.5 |
| 1 | 5.1 | 81.7 | 48.4 | 117.6 | 2.4 |
| 2 | 10.2 | 163.3 | 48.4 | 199.2 | 4.1 |
| 5 | 25.5 | 408.3 | 48.4 | 444.2 | 9.1 |
| 10 | 51 | 816.5 | 48.4 | 852.4 | 17.6 |





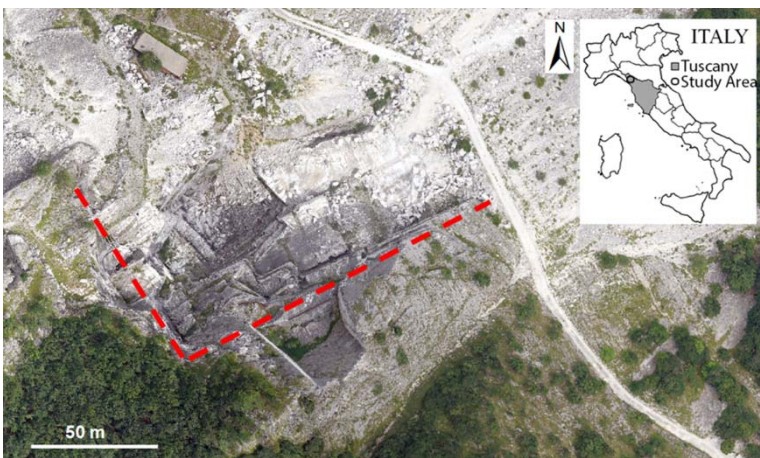

**Figure 1.** Top view of the Piastrone open pit with indication of the two principal slope directions (dotted lines). Inset map shows the location of the study area.





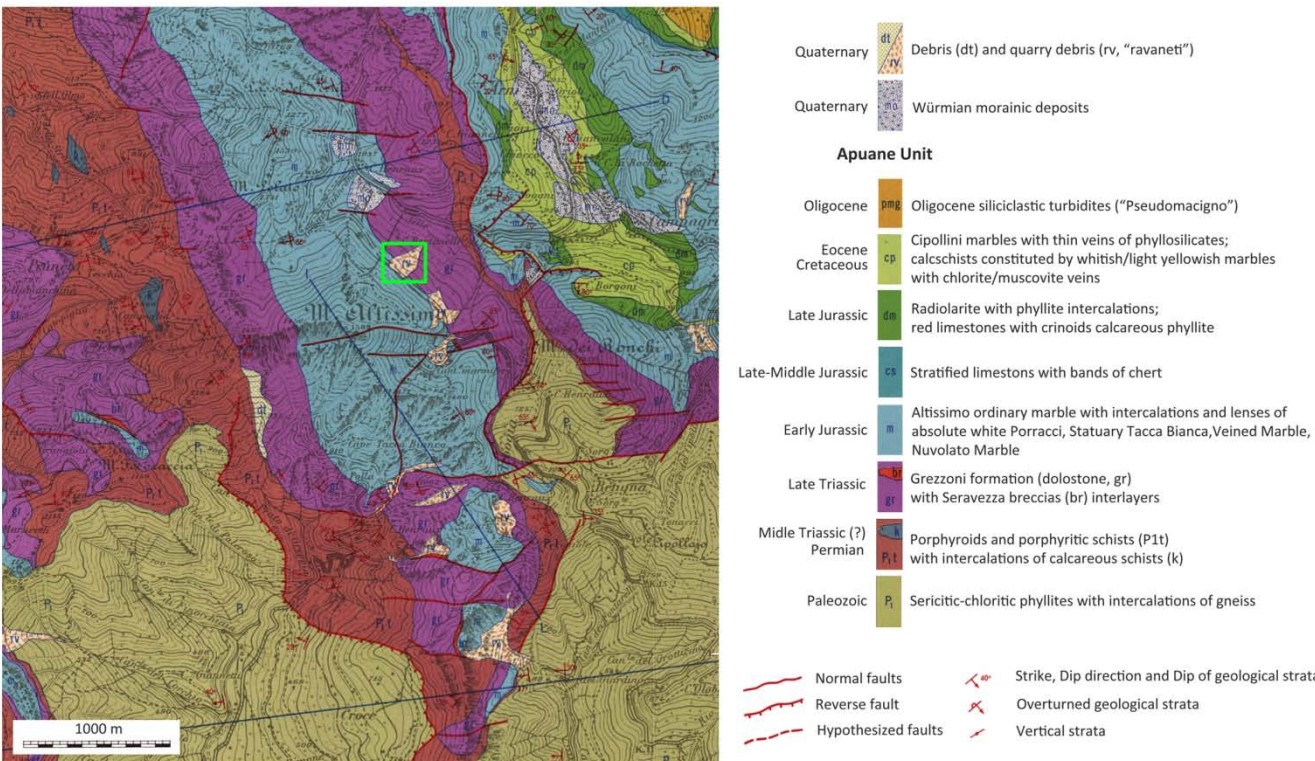

**Figure 2.** Geological map of the Mt. Altissimo area. The rectangle indicates the location of the quarry (modified from Giglia and Paiotti, 1963).





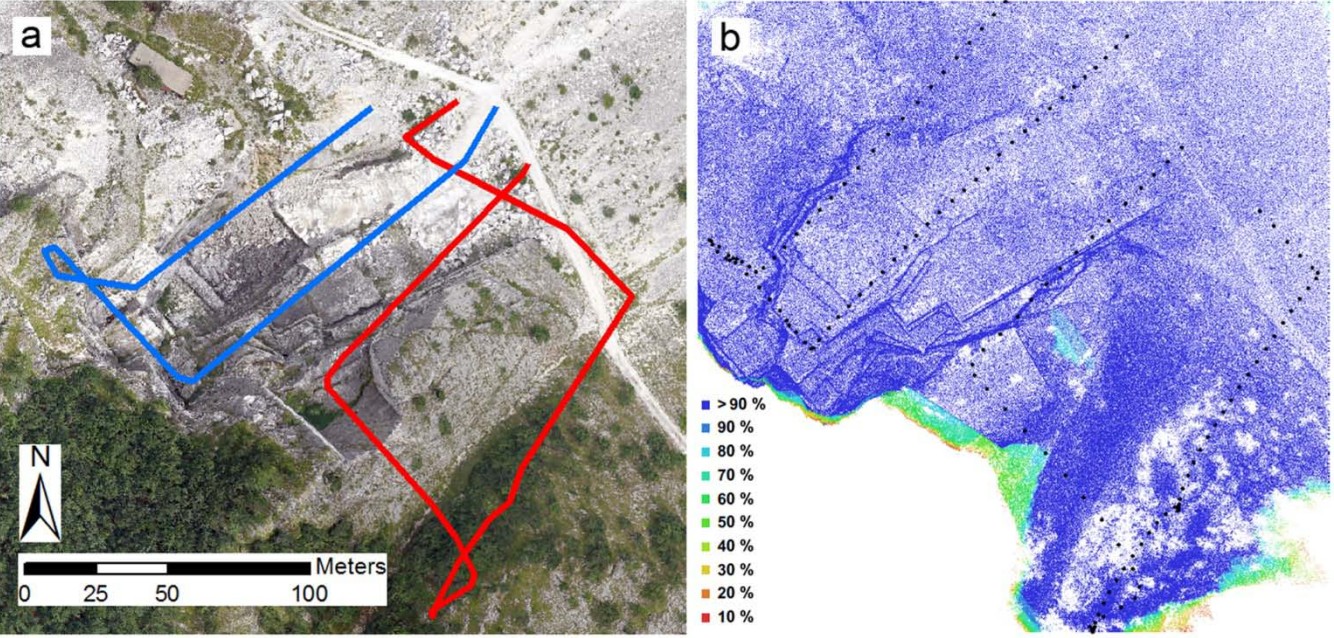

**Figure 3.** Flight paths of the RPAS zenithal surveys (a). Top view of the area with indication of camera locations (black dots) and image overlap percentage (b).





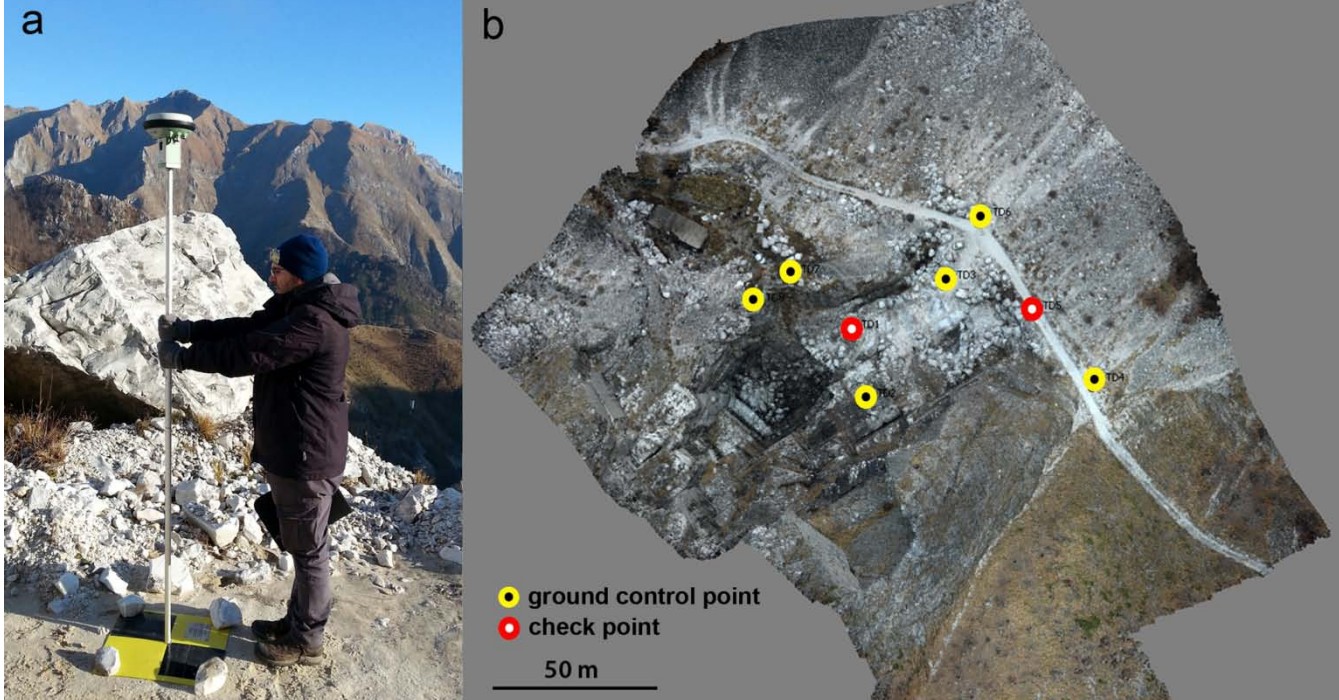

**Figure 4.** Example of a GCP measured during the RTK GNSS field survey (a) and spatial distribution of GCPs and check points for the RPAS zenithal flights (b).





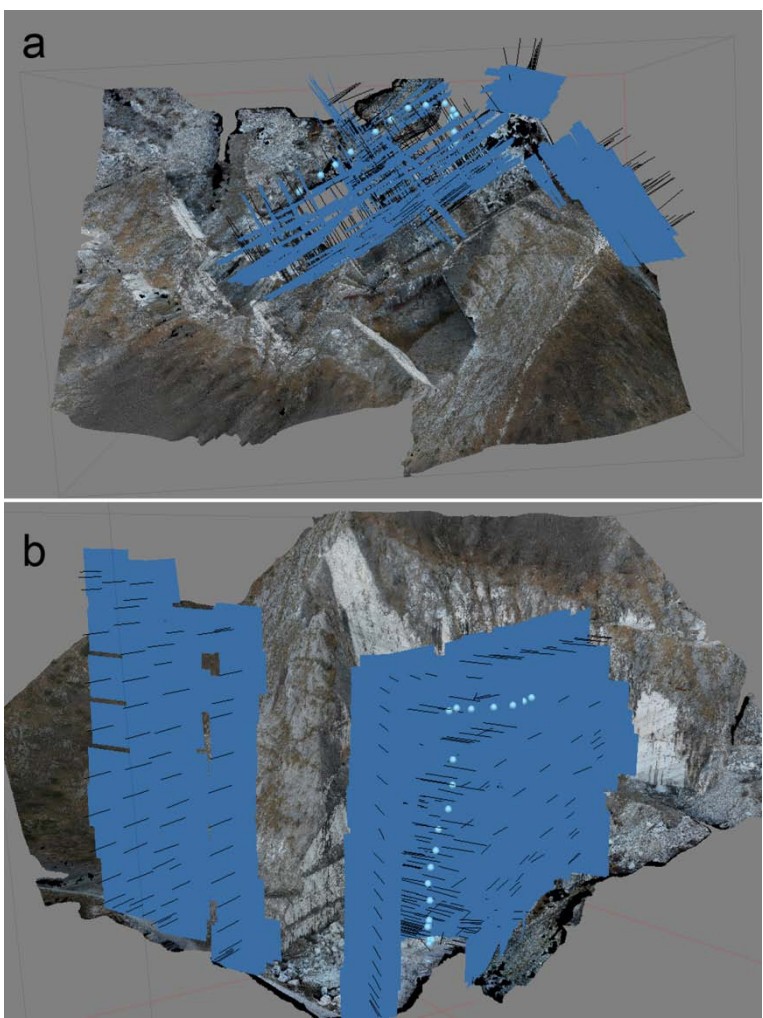

**Figure 5.** Top view (a) and perspective view (b) of the RPAS frontal surveys. Corresponding 3D point cloud produced with photogrammetric processing of digital images is shown in background.




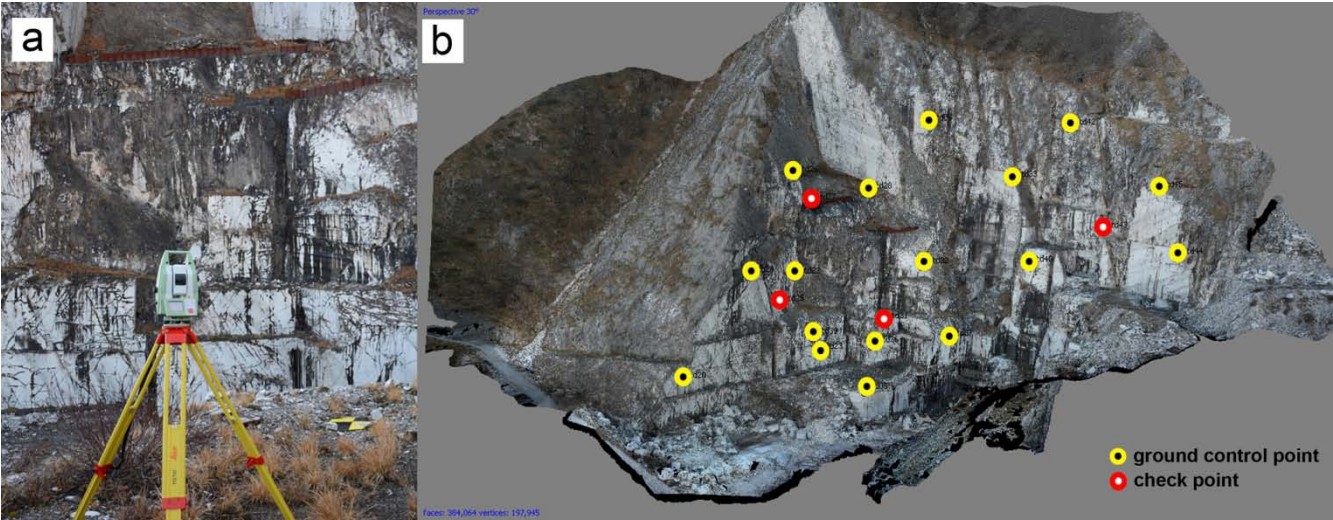

**Figure 6.** Topographic survey with TS (a) and spatial distribution of GCPs and check points for the RPAS frontal flights (b).





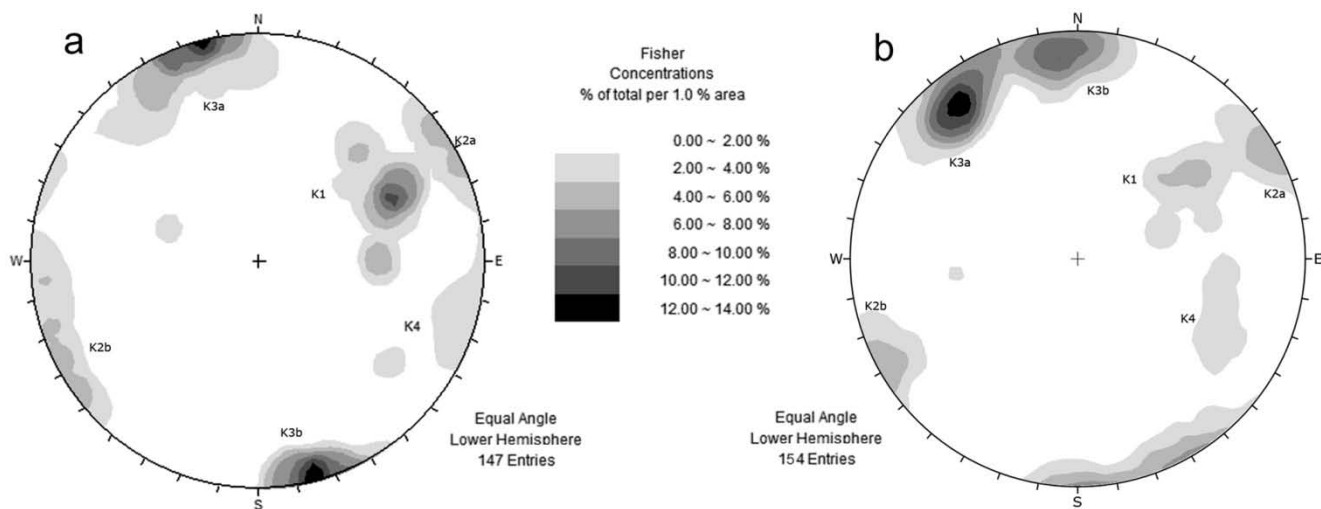

**Figure 7.** Stereonet plot of poles (Schmidt, equal area, lower hemisphere) of the discontinuities manually collected through classical engineering geological survey (a) and remotely collected by using photogrammetric data from RPAS surveys (b).





**Figure 8.** Examples of kinematic stability analysis carried out using Rocscience™ Dips 6.0 stereographic projection through the Wulff equal-angle method (lower hemisphere); a) Planar sliding on the eastern slope; b) Wedge sliding on the eastern slope; c) Direct toppling on the western slope; d) Aerial photo showing the two principal slope orientations.





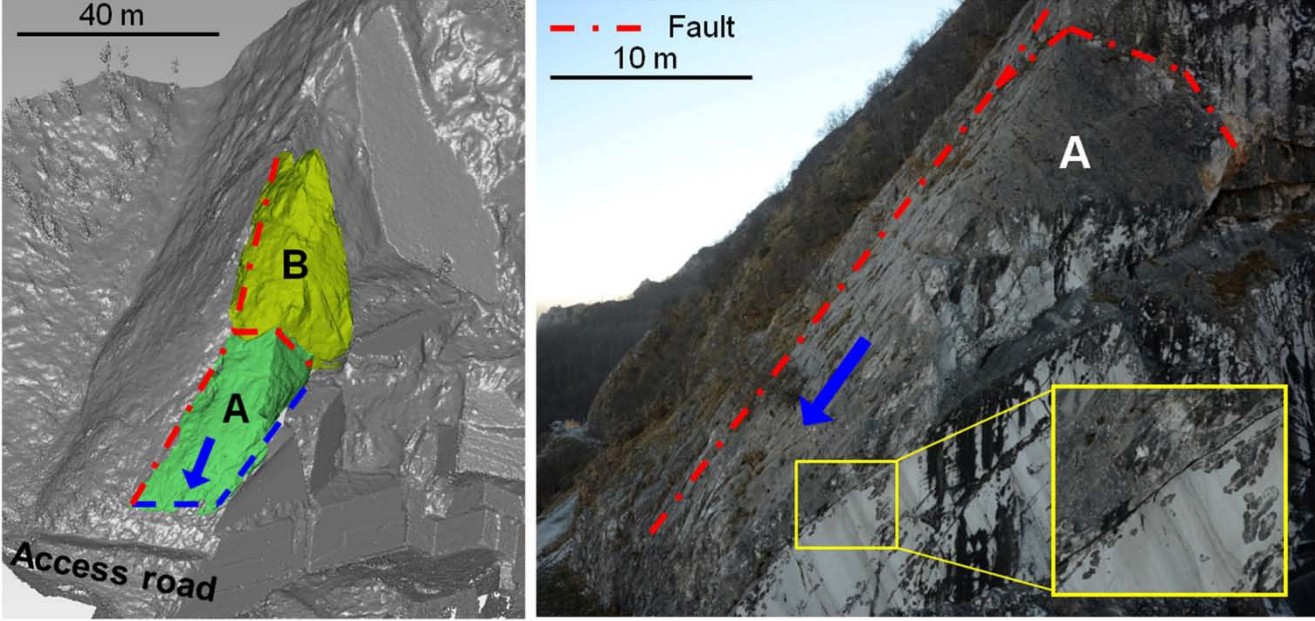

**Figure 9.** Identification of two large blocks with potential for sliding; insect photo highlights the visible aperture of the basal plane.





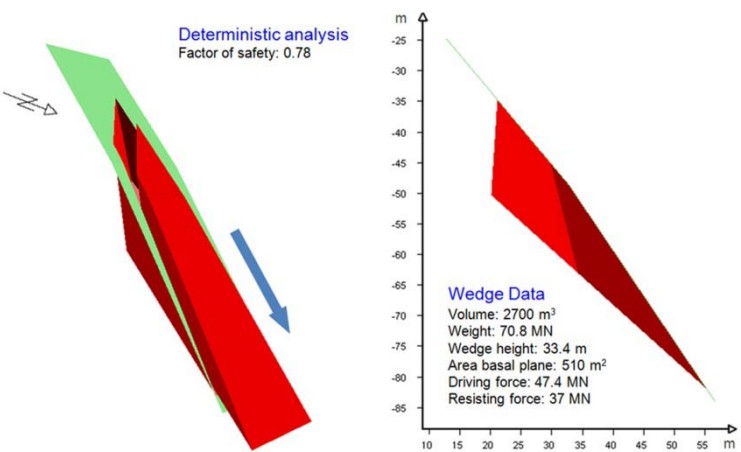

**Figure 10.** Result of Swedge 3D preliminary slope stability analysis.





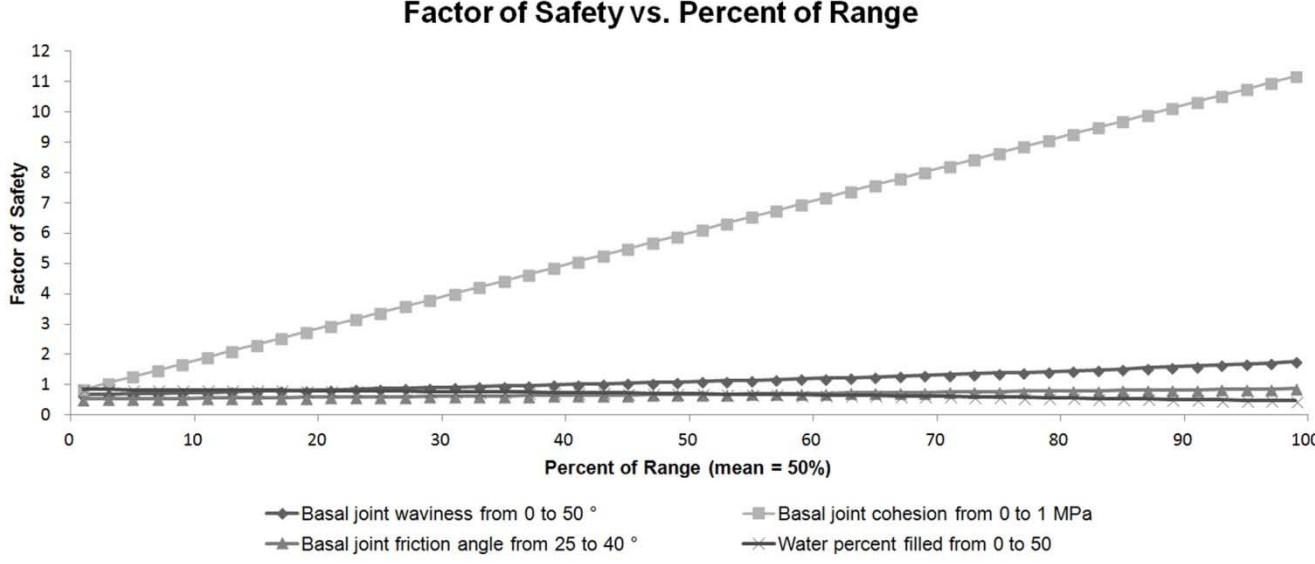

**Figure 11.** Result of sensitivity analysis relative to basal joint waviness, friction angle, cohesion and water percent filled parameters.





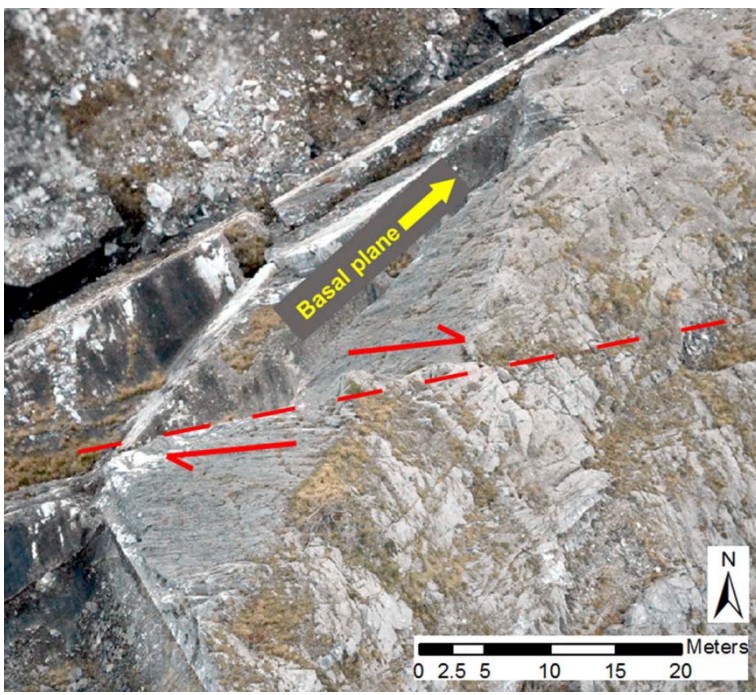

**Figure 12.** Particular of the zenithal orthophoto with indication of apparent motion of a major fault acting as back release surface for block A of figure 9.





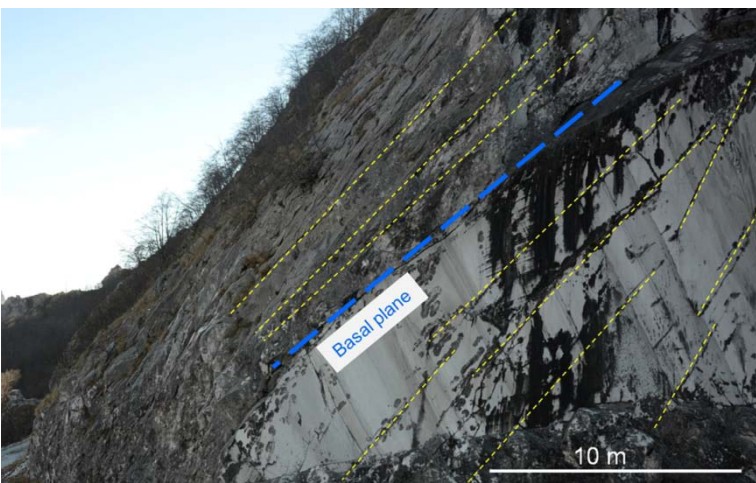

**Figure 13.** Particular of a series of thigh discontinuities over and below the basal plane under study.