# Peer review of "Use of a remotely piloted aircraft system for hazard assessment in a rocky mining area (Lucca, Italy)"

_Natural Hazards and Earth System Sciences, 2017_

## Referee Comment (RC1) · Anonymous Referee #1 · 5 Jul 2017

GENERAL COMMENTS:

This paper presents an interesting case study of the application of RPAS for rock slope characterization in a mine/quarry for hazard assessment. It highlights the advantages of using recently developed technologies (RPAS and SfM) in a mine/quarry.

In my opinion, the main contribution is related to the persistence of critical joints and the role of intact rock bridges in rock slope stability. This is a difficult topic, and lots of literature exists already. New characterization techniques, such as photogrammetry, bring new perspective and may allow better understanding of the role of rock bridges. Therefore, I think that this manuscript is a topical case study, and the discussion (Sec-

tion 5) is interesting.

However, before being published, I think the manuscript needs to be further completed. I would suggest including a more comprehensive literature review on the topic of discontinuity persistence and rock bridges in the introduction (including the current paragraph Line 14-19 on Page 8). I would suggest reviewing recent case studies on rock bridges such as the one by Frayssines and Hantz (2006), Sturzenegger and Stead (2012), Tuckey and Stead (2016), and Matasci et al (2014). In particular, the results presented in Line 15-18 on Page 9 could be compared to rock bridge percentage estimate by the above authors. Finally, the specific comments listed below should be addressed.

SPECIFIC COMMENTS:

Page 1, Title: Is there a specific reason why the authors use "remotely piloted aircraft system" instead of UAV, which is more commonly used in the literature?

The abstract is well written. I would suggest adding a sentence on the rock bridge analysis, which is an important aspect of this manuscript.

From Page 1, Line 33 to Page 2 Line 5: these sentences seem a bit vague. In what ways does alteration of geological structures by exploitation, or morphological features influence slope stability? In my opinion, the main parameter controlling slope stability is the relationship between the slope morphology and geological structures, as rightly explained in the third sentence.

Page 3, Lines 10-13: is it really necessary to add these sentences and to mention this accident? Safety is definitely very important for mining operation, but is this really relevant for the scientific contribution of this paper?

Section 3.1: Could "zenithal", "parallel" and "frontal" be defined?

Section 3.1: What is the exact meaning of Ground Sample Distance: is it the ground pixel size? Or the distance between points in the generated point clouds?

Section 3.2: I don't think it is necessary to explain in detail every steps of the processing work using Agisoft. It may be better to explain the key steps and refer to Agisoft manual for more information. Details about the parameters and options selected in Agisoft could be listed in a table if necessary. In addition, I would consider including Section 4.1 here instead of in the Result section of the manuscript.

Section 3.3, Lines 16-17 and Lines 25-26 do not seem necessary.

Page 7, Line 10: a table showing the parameters used to obtain the RMRb and GSI would be useful here. I assume the geometric parameters come from the RPAS, but what is the source of the non-geometric parameters?

Page 6, Line 30: is the reproduction error resulting only from manual placement of GCPs or also to other parameters of the alignment process?

Section 4.2: would it be possible to add a paragraph to discuss the results of the kinematic analysis? How do they compare with field-/SfM-based observations? What are the main failure mechanisms?

Page 7, Line 25: do the orientation of the faults and discontinuity basal plane correspond to specific discontinuity sets defined previously? I think the sentences Line 14-19 on Page 10 should appear here.

Section 4.3: it is not clear how Block A parameters shown in Table 5 were input in Swedge. What is the slope orientation? How were the geometric parameters of Table 5 used to generate the wedge shown in Figure 10? Can the "length" and "height" of the block in Table 5 be defined, or illustrated on Figure 9? How was Total Cohesion in Table 6 calculated?

TECHNICAL CORRECTIONS:

For clarity, I suggest subdividing the introduction into more paragraphs. I would start a new paragraph from (1) "Generally, ..." (Page 2, Line 5); (2) "However, ..." (Page 2, Line 13); (3) "Digital images..." (Page 2, Line 22); (4) "However, ..." (Page 2, Line 30),

and I suggest deleting the word "however" here.

I suggest starting a new paragraph on Page 10, Line 10 at "In this work"

All references to figures in the text should be in brackets (Fig. X)

Sections 3 and 4 need to be reviewed for clarity and the English checked.

Page 2, Line 3: I suggest using either "geological discontinuities" (Page 2, Line 3) or "geological structures" (Page 1, Line 4), but being consistent

Page 2, Line 6: I suggest adding a period and start a new sentence from "Measurement"

Page 2, Line 10: "DP" should be "TDP" for Terrestrial Digital Photogrammetry

Page 2, Line 12: "rocky outcrops" should be "rock outcrops". Similarly, on Page 7 "rocky slope" and "rocky blocks" should be "rock slope" and "rock block".

Page 2, Line 13: I suggest rephrasing this sentence, something like "A limitation of ground-based remote sensing is related to the survey of complex topography from suboptimal camera or scanner positions, resulting in occlusion zones..."

Page 2, line 16: I suggest deleting this sentence. It seems a bit redundant, and not really true, since the next sentences list several examples of the application of RPAS in open-pit mining.

Page 2, Line 20: delete "an"

Page 2, Line 26: a word is missing "...multicopters results ARE particularly suitable..."

Page 2, Line 28: delete "both"

Page 2, Line 34: should read " allow only a rough estimation of airborne camera external orientation"

Page 3, Line 1: I think the word "accurate" is not appropriate here, because SfM provide accurate models whether they are geo-referenced or not. I suggest rephrasing, something like "in order to geo-reference (or register) 3D models, ..."

Page 3, Line 3: should be "dependent not only ON" (not "from"); same comment at the end of the line

Page 3, Line 3: I suggest rephrasing and use "a preliminary rock fall hazard assessment, requested..."

Page 3, Line 24: I suggest rephrasing "The bottom of the pit is located at 1,180 meters above sea level (masl) and the top of the excavated rock face is at 1,300 masl.

Page 3, Line 29: "compressive tectonic phase WHICH originated..."

Page 3, Line 32: "fragile" should read "brittle"?

Page 4, Line 1: "motion" should read "displacement" or "offset"?

Page 4, Line 3-5: please rephrase with something like "AS involves the oldest LITHOLOGIES of the ..., INCLUDING pre-Alpine..."

Page 5, Line 5: would "baseline" be a better terminology for the "two points necessary for the roto-translation of the measured GCPs"?

Page 7, Line 20: where are the results of block shape and size? Do you mean to say that the results of the kinematic analysis highlight potential for discontinuity-controlled failure mechanism and "therefore the high resolution images and the dense point cloud were analyzed in order to locate possible block source areas"?

Page 7, Line 23: do you mean to say :"In particular, the adopted approach identified two large blocks..."?

Page 7, Line 29-30: I suggest moving this sentence to Line 25, after "high persistence".

Page 8, Line 14: I suggest wording "impossible to measure deterministically"

Page 8, Line 15: I suggest saying that for this reason, persistence is commonly measured as trace length on rock outcrop, and use a more appropriate reference than Einstein et al (1983)

Page 10, Line 10: "slope stability analysis" instead of "slope instability analysis"

Page 10, Line 25: the reference should be "(Kemeny and Donovan, 2005)"

Figures 3 and 5 captions: "top view" should read "plan view"

Figure 5 needs to be referenced in the text; the caption should explain that the blue rectangles correspond to the photographs locations; there is not scale nor indication of the north on the figure.

Figure 7: could you please clarify: the caption mention equal area, while the figure shows equal angle. In addition, Figure 7 uses Schmidt method while Figure 8 uses Wulff method.

Figure 9: "insect photo" should read "inset photo"

Figure 13 captions should read "Details of a series of tight discontinuities..."

---

## Referee Comment (RC2) · Anonymous Referee #2 · 13 Jul 2017

Dear Authors,

This paper shows not only survey results of complex morphologies using RPAS and SfM-MVS but also a practical application for disaster prevention using those high resolution data, therefore, very interesting. Since detailed measurement procedures, advantages and disadvantages of RPAS and SfM methods are also well explained, I think that this paper is worth to be published.

However additional explanations and reconsiderations for the following points should be desired.

Although high resolution 3 dimensional data were obtained using RPAS, does the

present stability analysis need that high resolution data? Since the higher resolution of data, the higher costs of data acquisition, processing and handling, appropriate resolution according to the purpose would exist.

Page 3, lines 10-13: Even though this paper deals with management of natural hazard, detailed description of a real victim would be not necessary in this paper discussing survey method and its application.

Figure 4: Although GCPs are located only in the bottom of cliff, is there any effect on the accuracy of 3D model of the cliff?

Figure 6: Although the number of GCPs looks too much, how did you decide their locations and number?

Yours sincerely,

---

## Author Comment (AC1) · 21 Aug 2017

We prepared a point by point response to review 1, please refer to the attached revised manuscript to see the applied changes.

RC -This paper presents an interesting case study of the application of RPAS for rock slope characterization in a mine/quarry for hazard assessment. It highlights the advantages of using recently developed technologies (RPAS and SfM) in a mine/quarry. In my opinion, the main contribution is related to the persistence of critical joints and the role of intact rock bridges in rock slope stability. This is a difficult topic, and lots of literature exists already. New characterization techniques, such as photogrammetry,

bring new perspective and may allow better understanding of the role of rock bridges. Therefore, I think that this manuscript is a topical case study, and the discussion (Section 5) is interesting. However, before being published, I think the manuscript needs to be further completed. I would suggest including a more comprehensive literature review on the topic of discontinuity persistence and rock bridges in the introduction (including the current paragraph Line 14-19 on Page 8). I would suggest reviewing recent case studies on rock bridges such as the one by Frayssines and Hantz (2006), Sturzenegger and Stead (2012), Tuckey and Stead (2016), and Matasci et al (2014). In particular, the results presented in Line 15-18 on Page 9 could be compared to rock bridge percentage estimate by the above authors.

AC -As suggested, a more comprehensive literature review on discontinuity persistence and rock bridges has been included in the introduction (including paragraph line 9-19 on page 8 and line 5-11 on page 9). The following text has been added: "Nevertheless, there are controlling factors that can have a great influence on the stability condition of a block or slope that cannot be fully determined, such as discontinuity persistence. The presence of intact rock bridges, that represent intervals of intact rock between adjacent discontinuities (ISRM, 1978), can significantly increase the stability of a rock slope, since the cohesion of the intact rock is generally of at least two orders of magnitude greater than the shear strength of a discontinuity (Park, 2005). In general, joint persistence (K) is defined as the fraction area that is actually discontinuous (Einstein et al., 1983), and can be calculated with the following Eq. (1):

$$K = \lim(A\_D \rightarrow \infty)((\sum\_i a\_Di)/A\_D) \quad (1)$$

where D is a region of the plane with area A_D and a_Di is the area of the joint in D. The limit of the application of this method is that the discontinuity area is practically impossible to measure deterministically in the field, for this reason persistence is commonly measured as trace length on rock outcrops. Jennings (1970) proposed the following Eq. (2) for persistence calculation starting from trace length values on rock exposure:

$$K = (\sum JL)/(\sum JL + \sum RBR) (2)$$

where JL is the total length of the joints segment and RBR is the total length of rock bridges. Mathematically, it is possible to consider the presence of rock bridges in terms of effective cohesion along the shear surface (Eberhardt et al., 2004) by using the following Eq. (3):

c_i=c A_g/A (3)

where c is the intact rock cohesion, A_g the total area of intact rock bridges along the shear surface, and A is the total area of the shear surface. Importantly, as recently reported by Tuckey and Stead (2016), in spite of the importance of intact rock bridges in slope stability, there are still no standard accepted methods for estimating the extent of rock bridges and incorporating rock bridges into slope stability analysis." In addition, the indicated case studies have been added in the discussion section, for purposing of comparison: "Similar values of rock bridges percentage have also been found in different case studies, where back-analysis revealed low values of estimated rock bridge content at the moment of failure, in the order if 0 to 5 % (Frayssines and Hantz, 2006; Grøneng et al., 2009; Sturzenegger and Stead, 2012; Matasci et al., 2014; Tuckey and Stead, 2016). Therefore, a small amount of rock bridge may be sufficient for guaranteeing stability of a rock slope."

RC -Finally, the specific comments listed below should be addressed. SPECIFIC COMMENTS: Page 1, Title: Is there a specific reason why the authors use "remotely piloted aircraft system" instead of UAV, which is more commonly used in the literature?

AC -The choice of using "remotely piloted aircraft system" instead of UAV is done trying to be as closer as possible to the title of Special Issue.

RC -The abstract is well written. I would suggest adding a sentence on the rock bridge analysis, which is an important aspect of this manuscript.

AC -We agree with the suggestion. The following sentence has been modified as it

follows: "A preliminary stability analysis, with focus on investigating the contribution of potential rock bridges, was then performed in order to demonstrate the potential use of RPAS information in engineering geological contexts for geo-hazard identification, awareness and reduction"

RC -From Page 1, Line 33 to Page 2 Line 5: these sentences seem a bit vague. In what ways does alteration of geological structures by exploitation, or morphological features influence slope stability? In my opinion, the main parameter controlling slope stability is the relationship between the slope morphology and geological structures, as rightly explained in the third sentence.

AC -The correction has been applied and the new sentences have been modified as it follows: " According to Zajc et al. (2014), for example, hazardous situations may occur when unfavourable sedimentological characteristics and geological discontinuities (e.g. joints, faults) of rock masses are made even more critical by extraction of the resource or ore material. In addition, Zheng et al. (2015) highlight the crucial role played by morphological features, such as sharp cuts and steep slopes, for potential triggering of rockfalls in mining areas. As widely demonstrated in the literature, the understanding of geometric relationships between geological discontinuities and slope morphology is essential to evaluate the potential occurrence of rock failures, since orientation of joint sets may influence both size and failure mechanisms of rock blocks prone to collapse (e.g. Stead and Wolter, 2015).".

RC -Page 3, Lines 10-13: is it really necessary to add these sentences and to mention this accident? Safety is definitely very important for mining operation, but is this really relevant for the scientific contribution of this paper?

AC -The suggestion has been applied and the sentences eliminated.

RC -Section 3.1: Could "zenithal", "parallel" and "frontal" be defined?

AC -The correction has been applied and the new sentences have been modified as it

follows: " In order to assess and localize the slope stability hazard in the rocky mining area, two RPAS surveys were carried out with direction of photo acquisition in zenithal modality (perpendicular to the open pit floor) and in frontal modality (perpendicular to the rock faces).".

RC -Section 3.1: What is the exact meaning of Ground Sample Distance: is it the ground pixel size? Or the distance between points in the generated point clouds?

AC -The sentence has been modified as it follows: "An average estimated distance between pixel centers measured on the ground (i.e. ground sample distance - GSD) of 2.4 cm was calculated.".

RC -Section 3.2: I don't think it is necessary to explain in detail every steps of the processing work using Agisoft. It may be better to explain the key steps and refer to Agisoft manual for more information. Details about the parameters and options selected in Agisoft could be listed in a table if necessary. In addition, I would consider including Section 4.1 here instead of in the Result section of the manuscript.

AC -Considering that the title of the Special Issue is "The use of remotely piloted aircraft systems in monitoring applications and management of natural hazards" we have considered this part very important in order to explain as better as possible the main steps of image processing and the utilized methods. In addition, we prefer to explain the utilized methods (ex. topographic survey with GPS and Total Station, GPS postprocessing, GCP correction to orthometric heights) more than just listing the parameters in a table. We consider Section 4.1 a description of the obtained results and not a method. For this reason we prefer to leave it as it is.

RC -Section 3.3, Lines 16-17 and Lines 25-26 do not seem necessary.

AC -We don't completely agree with these suggestions and we would prefer to leave them as they are, since they introduce the next sentences and steps of the analysis.

RC -Page 7, Line 10: a table showing the parameters used to obtain the RMRb and

GSI would be useful here. I assume the geometric parameters come from the RPAS, but what is the source of the non-geometric parameters?

AC -The non-geometric parameters were manually collected in accessible areas; we added a sentence in the text to explain this. Moreover, we added a table (table 4) where RMR parameters are shown. We didn't add a table for the GSI parameters since it is a "qualitative" index. We don't think it is useful to add an image with the GSI chart in this paper, anyhow we included in the text a comparison with the Hoek et al. (2013) equation. To summarize, the text was has been changed as follows: "The final stereonet allowed identification of four discontinuity sets, whose properties listed in table 3 were obtained from traditional engineering geological survey carried out in accessible areas of the mine.." "..Based on the discontinuity characteristics derived from RPAS and traditional engineering geological surveys, the basic RMR (RMRb) and GSI index were calculated. The RMRb was found to be 67 (table 4), while the GSI was estimated to be between 60 and 65 using the modified chart proposed by Hoek et al. (2013). In addition, application of Hoek et al. (2013) equation for GSI quantification (GSI=1.5 JCond89 + RQD/2) confirmed the results of the qualitative chart interpretation with a value of 65."

Please refer to the attached revised manuscript to see Table 4.

RC -Page 6, Line 30: is the reproduction error resulting only from manual placement of GCPs or also to other parameters of the alignment process?

AC -As described in Paragraph 3.2. the reprojection error does not result only from manual placement of GCPs but also from other parameters of the alignment process and from the characteristics of data acquisition. For these reasons the sentence has been modified as it follows: "The image alignment process, described in paragraph 3.2, resulted in a reprojection error of 0.41 pixel for the zenithal survey and 0.48 pixel for the frontal survey.".

RC -Section 4.2: would it be possible to add a paragraph to discuss the results of the

kinematic analysis? How do they compare with field-/SfM-based observations? What are the main failure mechanisms?

AC -We added the following sentence: "Three different possible kinematic modes were identified, with K2b and K4 systems having the most influence on potential instability. The majority of the potential failures identified relate to planar sliding or wedge sliding, in agreement with field and SfM-based observations."

RC -Page 7, Line 25: do the orientation of the faults and discontinuity basal plane correspond to specific discontinuity sets defined previously? I think the sentences Line 14-19 on Page 10 should appear here.

AC -We agree with the suggestion. Therefore, the following sentence was included where indicated: "The basal plane appears not to correspond with any of the identified discontinuity sets, but is probably connected to planes of weakness of the marble in correspondence with a particular orientation of minerals crystallographic axes. The lateral and rear faults, however, may be associated with the K3a and K3b systems respectively. The rear fault may also be associated with the East-West fault system that characterizes the geology of this area of the Apuan Alps complex (Fig. 2)."

RC -Section 4.3: it is not clear how Block A parameters shown in Table 5 were input in Swedge. What is the slope orientation? How were the geometric parameters of Table 5 used to generate the wedge shown in Figure 10? Can the "length" and "height" of the block in Table 5 be defined, or illustrated on Figure 9? How was Total Cohesion in Table 6 calculated?

AC -In Swedge the block can be created using few input data (Slope Dip/Dip Direction and Height; rear discontinuity Dip and Dip Direction; lateral and basal plane Dip/Dip Directions). We think that including explanation on how to insert data into Swedge is not necessary in this case. Moreover, the geometric characteristic of the block can be derived from bar scale of figure 9 and vertical and horizontal axes on figure 10. Nevertheless we added a sentence specifying the slope direction used in the analysis: "The

geometry of Block A was deterministically re-created in Swedge using the geometrical information obtained from the point cloud provided in table 6, with a slope direction of 30 degrees.". Concerning the cohesion values on table 6, they were calculated according to equation (3) c_i=cA_g/A. The equation allows calculation of effective cohesion due to rock bridges. For example, if we consider 1 m2 discontinuity plane with the 2 % of rock bridges, the effective cohesion of that plane will be equal to 16 MPa (intact rock cohesion) times 0,02/1 that is 0.32 MPa (effective cohesion considering 2% of rock bridge). This is the value to be used in Swedge. However, in our case study the basal plane is 510 m2, therefore the total effective cohesion is 0.32 times 510, that is 163.3 MN. The driving force on table 6 instead, is mainly due to the effect of the block weight on the basal plane. In our opinion all these information are already present in the text, nevertheless we are willing to discuss eventual modification if needed.

RC -TECHNICAL CORRECTIONS: For clarity, I suggest subdividing the introduction into more paragraphs. I would start a new paragraph from (1) "Generally, ..." (Page 2, Line 5); (2) "However, ..." (Page 2, Line 13); (3) "Digital images..." (Page 2, Line 22); (4) "However, ..." (Page 2, Line 30), and I suggest deleting the word "however" here.

AC -The suggestions have been applied

RC -I suggest starting a new paragraph on Page 10, Line 10 at "In this work"

AC -The suggestion has been applied

RC -All references to figures in the text should be in brackets (Fig. X)

AC -The correction has been applied

RC -Sections 3 and 4 need to be reviewed for clarity and the English checked.

AC -The text has been checked by a native English speaker and corrected where needed. Please, refer to the attached revised manuscript to see the applied changes.

RC -Page 2, Line 3: I suggest using either "geological discontinuities" (Page 2, Line 3)

[Figure]

or "geological structures" (Page 1, Line 4), but being consistent

AC -The suggestion has been applied

RC -Page 2, Line 6: I suggest adding a period and start a new sentence from "Measurement"

AC -The suggestion has been applied

RC -Page 2, Line 10: "DP" should be "TDP" for Terrestrial Digital Photogrammetry

AC -The correction has been applied

RC -Page 2, Line 12: "rocky outcrops" should be "rock outcrops". Similarly, on Page 7 "rocky slope" and "rocky blocks" should be "rock slope" and "rock block".

AC -The corrections have been applied

RC -Page 2, Line 13: I suggest rephrasing this sentence, something like "A limitation of ground-based remote sensing is related to the survey of complex topography from suboptimal camera or scanner positions, resulting in occlusion zones..."

AC -The correction has been applied

RC -Page 2, line 16: I suggest deleting this sentence. It seems a bit redundant, and not really true, since the next sentences list several examples of the application of RPAS in open-pit mining.

AC -The correction has been applied and the new sentence has been modified as it follows: "There are several photogrammetric studies using RPAS for the geomorphic feature characterization or mapping of the surface extent in open-pit mines (Lamb, 2000; Chen et al., 2015; Shahbazi et al., 2015; Tong et al., 2015; Esposito et al., 2017). Few of them concern the use of RPAS for discontinuity characterization of rock slopes affected by mining activity.....".

RC -Page 2, Line 20: delete "an"

AC -The correction has been applied

RC -Page 2, Line 26: a word is missing "...multicopters results ARE particularly suitable..."

AC -The correction has been applied and the new sentence has been modified as it follows: "In order to analyze rock outcrops, the use of RPAS multicopters results particularly suitable because it allows different geometric configurations for the image acquisition (i.e. zenithal, frontal, oblique).".

RC -Page 2, Line 28: delete "both"

AC -The correction has been applied

RC -Page 2, Line 34: should read " allow only a rough estimation of airborne camera external orientation"

AC -The correction has been applied

RC -Page 3, Line 1: I think the word "accurate" is not appropriate here, because SfM provide accurate models whether they are geo-referenced or not. I suggest rephrasing, something like "in order to geo-reference (or register) 3D models, ..."

AC -The correction has been applied and the new sentence has been modified as it follows: "In order to obtain accurate and georeferenced the 3D models, the use of ground control points (GCPs) surveyed with geodetic GNSS receivers and total station (TS) is generally employed (Francioni et al. 2015).".

RC -Page 3, Line 3: should be "dependent not only ON" (not "from"); same comment at the end of the line

AC -The correction has been applied

RC -Page 3, Line 3: I suggest rephrasing and use "a preliminary rock fall hazard assessment, requested..."

AC -The correction has been applied

RC -Page 3, Line 24: I suggest rephrasing "The bottom of the pit is located at 1,180 meters above sea level (masl) and the top of the excavated rock face is at 1,300 masl.

AC -The correction has been applied and the new sentence has been modified as it follows: "The bottom of the pit is located at 1,180 meters above sea level (m.a.s.l.) and the top of the excavated rock face is at 1,300 m.a.s.l..".

RC -Page 3, Line 29: "compressive tectonic phase WHICH originated..."

AC -The correction has been applied

RC -Page 3, Line 32: "fragile" should read "brittle"?

AC -The correction has been applied

RC -Page 4, Line 1: "motion" should read "displacement" or "offset"?

AC -The correction has been applied and the word "displacement" has been used instead of "motion".

RC -Page 4, Line 3-5: please rephrase with something like "AS involves the oldest LITHOLOGIES of the ..., INCLUDING pre-Alpine..."

AC -The correction has been applied

RC -Page 5, Line 5: would "baseline" be a better terminology for the "two points necessary for the roto-translation of the measured GCPs"?

AC -The correction has not been applied since the terminology "baseline" is correct, especially for GPS measurements, but, in our opinion, less explicative than our long sentence to explain the concept of roto-translation.

RC -Page 7, Line 20: where are the results of block shape and size? Do you mean to say that the results of the kinematic analysis highlight potential for discontinuity-controlled failure mechanism and "therefore the high resolution images and the dense

point cloud were analyzed in order to locate possible block source areas"?

AC -We wanted to say that since the traditional kinematic analyses don't allow local-ization of blocks source areas, the high resolution images were used to localize them. We don't think it is necessary to include the properties of each block, since the pa-per focus on the 2 bigger and most dangerous blocks. Nevertheless, the sentence wasn't clear, and it has been rewritten it in this way: "The results highlight the potential for blocks to form that may be subject to gravity induced instability but, as previously stated, traditional kinematic analyses do not identify the location of these unstable blocks. Therefore, further analysis of the high resolution images and the dense point cloud was performed in order to locate possible block source areas. More than 20 blocks were deterministically characterized in terms of size, shape and barycentric co-ordinates, varying from about a cubic meter to a few hundred cubic meters."

RC -Page 7, Line 23: do you mean to say :"In particular, the adopted approach identi-fied two large blocks..."?

AC -The correction has been applied.

RC -Page 7, Line 29-30: I suggest moving this sentence to Line 25, after "high persis-tence".

AC -The correction has been applied.

RC -Page 8, Line 14: I suggest wording "impossible to measure deterministically"

AC -The correction has been applied.

RC -Page 8, Line 15: I suggest saying that for this reason, persistence is commonly measured as trace length on rock outcrop, and use a more appropriate reference than Einstein et al (1983)

AC -The sentence was rewritten in this way: "The limit of the application of this method is that the discontinuity area is practically impossible to measure deterministically in

the field, for this reason persistence is commonly measured as trace length on rock outcrops. Jennings (1970) proposed the following Eq. (2) for persistence calculation starting from trace length values on rock exposure:"

RC -Page 10, Line 10: "slope stability analysis" instead of "slope instability analysis"

AC -The correction has been applied and "instability analysis" has been changed in "stability analysis" in the whole text.

RC -Page 10, Line 25: the reference should be "(Kemeny and Donovan, 2005)"

AC -The correction has been applied.

RC -Figures 3 and 5 captions: "top view" should read "plan view"

AC -The correction has been applied. Figure1 has been similarly modified.

RC -Figure 5 needs to be referenced in the text; the caption should explain that the blue rectangles correspond to the photographs locations; there is not scale nor indication of the north on the figure.

AC -Figure 5 was already referenced in the text (Pag. 5, Line 2). The caption has been modified and the sentence "blue rectangles correspond to the photographs locations, black lines to normals" has been added. A reference scale and the indication of the north have been added to the Figure 5a. Please, refer to the attached revised manuscript to see the modified figure.

RC -Figure 7: could you please clarify: the caption mention equal area, while the figure shows equal angle. In addition, Figure 7 uses Schmidt method while Figure 8 uses Wulff method.

AC -That was a mistake, the figure has been changed and "Equal angle" corrected. Please, refer to the attached revised manuscript to see the modified figure. Figure 7 uses Schmidt method since it represents a joint density analysis, while Figure 8 uses Wulff method since it refers to a slope kinematic stability analysis. They are both in

theory correct.

RC -Figure 9: "insect photo" should read "inset photo"

AC -The correction has been applied

RC -Figure 13 captions should read "Details of a series of tight discontinuities..."

AC -The correction has been applied

Please also note the supplement to this comment:
https://www.nat-hazards-earth-syst-sci-discuss.net/nhess-2017-194/nhess-2017-194-AC1-supplement.pdf

**Supplement:**

[revised manuscript text omitted]

---

## Author Comment (AC2) · 21 Aug 2017

We prepared a point by point response to referee 2.

RC -Dear Authors, This paper shows not only survey results of complex morphologies using RPAS and SfM-MVS but also a practical application for disaster prevention using those high resolution data, therefore, very interesting. Since detailed measurement procedures, advantages and disadvantages of RPAS and SfM methods are also well explained, I think that this paper is worth to be published. However additional expla-nations and reconsiderations for the following points should be desired. Although high resolution 3 dimensional data were obtained using RPAS, does the present stability

analysis need that high resolution data? Since the higher resolution of data, the higher costs of data acquisition, processing and handling, appropriate resolution according to the purpose would exist.

AC -We consider high resolution of data always useful in slope stability analyses because it allows the identification and measurement, with high precision and detail, of joints and potential unstable blocks and rock masses at any height above the open pit floor. As written in the Acknowledgments section, that part of the present study has been undertaken within the framework of an agreement with USL1 of Massa and Carrara (Mining Engineering Operative Unit - Department of Prevention) aimed to that purposes. Furthermore, high detail and accurate geometrical data allow deterministic kinematic analyses and the creation of reliable stability models. Costs of data acquisition, processing and handling are not a problem if compared with that of slope safety and risk reduction for workers.

RC -Page 3, lines 10-13: Even though this paper deals with management of natural hazard, detailed description of a real victim would be not necessary in this paper discussing survey method and its application.

AC -The suggestion has been applied and the sentences eliminated.

RC -Figure 4: Although GCPs are located only in the bottom of cliff, is there any effect on the accuracy of 3D model of the cliff?

AC -Although GCPs of the zenithal flight are located at the bottom of cliff, there is not any effect on the accuracy of 3D model of the cliff because they are well spatially distributed, redundant, the flight altitude is low and some photos are inevitably convergent; in particular, this last characteristic allowed us to build an accurate 3D model even in the surroundings of vertical quarry walls. Furthermore, it must be considered that frontal flights, on the perpendicular to the rock faces, have been executed ad hoc and the related photos used to build a separate frontal model as shown in Figure 6.

RC -Figure 6: Although the number of GCPs looks too much, how did you decide their locations and number?

AC -We decided to measure a great number of GCPs (21) because we had to orient, as more accurate as possible, 448 images.... GCPs location has been decided in a way to have an optimum spatial distribution (Figure 6) both in space, considering the V shape of the "Piastrone" quarry, and in elevation from the open pit floor.

---

## Author Response (AR1)

**Author's Response - NHESS-2017-194 Salvini et al. "Use of a remotely piloted aircraft system for hazard assessment in a rocky mining area (Lucca, Italy)"**

**Comment to Reviewers**

**Point-by-point response to Reviewer #1**
GENERAL COMMENTS:
This paper presents an interesting case study of the application of RPAS for rock slope characterization in a mine/quarry for hazard assessment. It highlights the advantages of using recently developed technologies (RPAS and SfM) in a mine/quarry.
In my opinion, the main contribution is related to the persistence of critical joints and the role of intact rock bridges in rock slope stability. This is a difficult topic, and lots of literature exists already. New characterization techniques, such as photogrammetry, bring new perspective and may allow better understanding of the role of rock bridges.
Therefore, I think that this manuscript is a topical case study, and the discussion (Section 5) is interesting.
However, before being published, I think the manuscript needs to be further completed.
I would suggest including a more comprehensive literature review on the topic of discontinuity persistence and rock bridges in the introduction (including the current paragraph Line 14-19 on Page 8). I would suggest reviewing recent case studies on rock bridges such as the one by Frayssines and Hantz (2006), Sturzenegger and Stead (2012), Tuckey and Stead (2016), and Matasci et al (2014). In particular, the results presented in Line 15-18 on Page 9 could be compared to rock bridge percentage estimate by the above authors.
As suggested, a more comprehensive literature review on discontinuity persistence and rock bridges has been included in the introduction (including paragraph line 9-19 on page 8 and line 5-11 on page 9). The following text has been added: "Nevertheless, there are controlling factors that can have a great influence on the stability condition of a block or slope that cannot be fully determined, such as discontinuity persistence. The presence of intact rock bridges, that represent intervals of intact rock between adjacent discontinuities (ISRM, 1978), can significantly increase the stability of a rock slope, since the cohesion of the intact rock is generally of at least two orders of magnitude greater than the shear strength of a discontinuity (Park, 2005). In general, joint persistence (K) is defined as the fraction area that is actually discontinuous (Einstein et al., 1983), and can be calculated with the following Eq. (1):

$$K = \lim_{A_D \to \infty} \frac{\sum_i a_{Di}}{A_D} \quad (1)$$

where D is a region of the plane with area AD and aDi is the area of the joint in D.
The limit of the application of this method is that the discontinuity area is practically impossible to measure deterministically in the field, for this reason persistence is commonly measured as trace length on rock outcrops. Jennings (1970) proposed the following Eq. (2) for persistence calculation starting from trace length values on rock exposure:

$$K = \frac{\sum JL}{\sum JL + \sum RBR} \quad (2)$$

where JL is the total length of the joints segment and RBR is the total length of rock bridges.
Mathematically, it is possible to consider the presence of rock bridges in terms of effective cohesion along the shear surface (Eberhardt et al., 2004) by using the following Eq. (3):

$$c_i = c \frac{A_g}{A} \quad (3)$$

where c is the intact rock cohesion, Ag the total area of intact rock bridges along the shear surface, and A is the total area of the shear surface.
Importantly, as recently reported by Tuckey and Stead (2016), in spite of the importance of intact rock bridges in slope stability, there are still no standard accepted methods for estimating the extent of rock bridges and incorporating rock bridges into slope stability analysis."
In addition, the indicated case studies have been added in the discussion section, for purposing of comparison: "Similar values of rock bridges percentage have also been found in different case studies, where back-analysis revealed low values of estimated rock bridge content at the moment of failure, in the order if 0 to 5 % (Frayssines and Hantz, 2006; Grøneng et al., 2009; Sturzenegger and Stead, 2012; Matasci et al., 2014; Tuckey and Stead, 2016). Therefore, a small amount of rock bridge may be sufficient for guaranteeing stability of a rock slope."

Finally, the specific comments listed below should be addressed.

SPECIFIC COMMENTS:
Page 1, Title: Is there a specific reason why the authors use "remotely piloted aircraft system" instead of UAV, which is more commonly used in the literature?
The choice of using "remotely piloted aircraft system" instead of UAV is done trying to be as closer as possible to the title of Special Issue.

The abstract is well written. I would suggest adding a sentence on the rock bridge analysis, which is an important aspect of this manuscript.
We agree with the suggestion. The following sentence has been modified as it follows: "A preliminary stability analysis, with focus on investigating the contribution of potential rock bridges, was then performed in order to demonstrate the potential use of RPAS information in engineering geological contexts for geo-hazard identification, awareness and reduction"

From Page 1, Line 33 to Page 2 Line 5: these sentences seem a bit vague. In what ways does alteration of geological structures by exploitation, or morphological features influence slope stability? In my opinion, the main parameter controlling slope stability is the relationship between the slope morphology and geological structures, as rightly explained in the third sentence.
The correction has been applied and the new sentences have been modified as it follows: " According to Zajc et al. (2014), for example, hazardous situations may occur when unfavourable sedimentological characteristics and geological discontinuities (e.g. joints, faults) of rock masses are made even more critical by extraction of the resource or ore material. In addition, Zheng et al. (2015) highlight the crucial role played by morphological features, such as sharp cuts and steep slopes, for potential triggering of rockfalls in mining areas. As widely demonstrated in the literature, the understanding of geometric relationships between geological discontinuities and slope morphology is essential to evaluate the potential occurrence of rock failures, since orientation of joint sets may influence both size and failure mechanisms of rock blocks prone to collapse (e.g. Stead and Wolter, 2015).".

Page 3, Lines 10-13: is it really necessary to add these sentences and to mention this accident? Safety is definitely very important for mining operation, but is this really relevant for the scientific contribution of this paper?
The suggestion has been applied and the sentences eliminated.

Section 3.1: Could "zenithal", "parallel" and "frontal" be defined?
The correction has been applied and the new sentences have been modified as it follows: " In order to assess and localize the slope stability hazard in the rocky mining area, two RPAS surveys were carried out with direction of photo acquisition in zenithal modality (perpendicular to the open pit floor) and in frontal modality (perpendicular to the rock faces).".

Section 3.1: What is the exact meaning of Ground Sample Distance: is it the ground pixel size? Or the distance between points in the generated point clouds?
The sentence has been modified as it follows: "An average estimated distance between pixel centers measured on the ground (i.e. ground sample distance - GSD) of 2.4 cm was calculated.".

Section 3.2: I don't think it is necessary to explain in detail every steps of the processing work using Agisoft. It may be better to explain the key steps and refer to Agisoft manual for more information. Details about the parameters and options selected in Agisoft could be listed in a table if necessary. In addition, I would consider including Section 4.1 here instead of in the Result section of the manuscript.
Considering that the title of the Special Issue is "The use of remotely piloted aircraft systems in monitoring applications and management of natural hazards" we have considered this part very important in order to explain as better as possible the main steps of image processing and the utilized methods. In addition, we prefer to explain the utilized methods (ex. topographic survey with GPS and Total Station, GPS post-processing, GCP correction to orthometric heights) more than just listing the parameters in a table. We consider Section 4.1 a description of the obtained results and not a method. For this reason we prefer to leave it as it is.

Section 3.3, Lines 16-17 and Lines 25-26 do not seem necessary.

We don't completely agree with these suggestions and we would prefer to leave them as they are, since they introduce the next sentences and steps of the analysis.

Page 7, Line 10: a table showing the parameters used to obtain the RMRb and GSI would be useful here. I assume the geometric parameters come from the RPAS, but what is the source of the non-geometric parameters?

The non-geometric parameters were manually collected in accessible areas; we added a sentence in the text to explain this. Moreover, we added a table (table 4) where RMR parameters are shown. We didn't add a table for the GSI parameters since it is a "qualitative" index. We don't think it is useful to add an image with the GSI chart in this paper, anyhow we included in the text a comparison with the Hoek et al. (2013) equation. To summarize, the text was has been changed as follows: "The final stereonet allowed identification of four discontinuity sets, whose properties listed in table 3 were obtained from traditional engineering geological survey carried out in accessible areas of the mine.." "..Based on the discontinuity characteristics derived from RPAS and traditional engineering geological surveys, the basic RMR (RMRb) and GSI index were calculated. The RMRb was found to be 67 (table 4), while the GSI was estimated to be between 60 and 65 using the modified chart proposed by Hoek et al. (2013). In addition, application of Hoek et al. (2013) equation for GSI quantification (GSI=1.5 JCond89 + RQD/2) confirmed the results of the qualitative chart interpretation with a value of 65."

**Table 4.** Parameters used for RMRb determination. Chosen index values are underlined.

| | Parameter | K1 | K2a | K2b | K3a | K3b | K4 |
|---|---|---|---|---|---|---|---|
| A1 | Strength of intact rock material | | | 7 | | | |
| A2 | RQD | | | (75%) 17 | | | |
| A3 | Spacing of discontinuities | 15 | 20 | 20 | 20 | 20 | 15 |
| A4 | Condition of discontinuities | 19 | 19 | 19 | 18 | 18 | 20 |
| A5 | Groundwater | 10 | 15 | 15 | 10 | 10 | 10 |
| | **RMRb 67** | | | | | | |

Page 6, Line 30: is the reproduction error resulting only from manual placement of GCPs or also to other parameters of the alignment process?

As described in Paragraph 3.2. the reprojection error does not result only from manual placement of GCPs but also from other parameters of the alignment process and from the characteristics of data acquisition. For these reasons the sentence has been modified as it follows: "The image alignment process, described in paragraph 3.2, resulted in a reprojection error of 0.41 pixel for the zenithal survey and 0.48 pixel for the frontal survey.".

Section 4.2: would it be possible to add a paragraph to discuss the results of the kinematic analysis? How do they compare with field-/SfM-based observations? What are the main failure mechanisms?

We added the following sentence: "Three different possible kinematic modes were identified, with K2b and K4 systems having the most influence on potential instability. The majority of the potential failures identified relate to planar sliding or wedge sliding, in agreement with field and SfM-based observations."

Page 7, Line 25: do the orientation of the faults and discontinuity basal plane correspond to specific discontinuity sets defined previously? I think the sentences Line 14-19 on Page 10 should appear here.

We agree with the suggestion. Therefore, the following sentence was included where indicated: "The basal plane appears not to correspond with any of the identified discontinuity sets, but is probably connected to planes of weakness of the marble in correspondence with a particular orientation of minerals crystallographic axes. The lateral and rear faults, however, may be associated with the K3a and K3b systems respectively. The rear fault may also be associated with the East-West fault system that characterizes the geology of this area of the Apuan Alps complex (Fig. 2)."

Section 4.3: it is not clear how Block A parameters shown in Table 5 were input in Swedge. What is the slope orientation? How were the geometric parameters of Table 5 used to generate the wedge shown in Figure 10? Can the "length" and "height" of the block in Table 5 be defined, or illustrated on Figure 9? How was Total Cohesion in Table 6 calculated?

In Swedge the block can be created using few input data (see image below).

[Figure]

We think that including explanation on how to insert data into Swedge is not necessary in this case. Moreover, the geometric characteristic of the block can be derived from bar scale of figure 9 and vertical and horizontal axes on figure 10. Nevertheless we added a sentence specifying the slope direction used in the analysis: "The geometry of Block A was deterministically re-created in Swedge using the geometrical information obtained from the point cloud provided in table 6, with a slope direction of 30 degrees.".

Concerning the cohesion values on table 6, they were calculated according to equation (3) $c_i = c\frac{A_g}{A}$.

The equation allows calculation of effective cohesion due to rock bridges. For example, if we consider 1 m² discontinuity plane with the 2 % of rock bridges, the effective cohesion of that plane will be equal to 16 MPa (intact rock cohesion) times 0,02/1 that is 0.32 MPa (effective cohesion considering 2% of rock bridge). This is the value to be used in Swedge. However, in our case study the basal plane is 510 m², therefore the total effective cohesion is 0.32 times 510, that is 163.3 MN. The driving force on table 6 instead, is mainly due to the effect of the block weight on the basal plane. In our opinion all these information are already present in the text, nevertheless we are willing to discuss eventual modification if needed.

TECHNICAL CORRECTIONS:
For clarity, I suggest subdividing the introduction into more paragraphs. I would start a new paragraph from (1) "Generally, ..." (Page 2, Line 5); (2) "However, ..." (Page 2, Line 13); (3) "Digital images..." (Page 2, Line 22); (4) "However, ..." (Page 2, Line 30), and I suggest deleting the word "however" here.
The suggestions have been applied

I suggest starting a new paragraph on Page 10, Line 10 at "In this work"
The suggestion has been applied

All references to figures in the text should be in brackets (Fig. X)
The correction has been applied

Sections 3 and 4 need to be reviewed for clarity and the English checked.
The text has been checked by a native English speaker and corrected where needed.

Page 2, Line 3: I suggest using either "geological discontinuities" (Page 2, Line 3) or "geological structures" (Page 1, Line 4), but being consistent
The suggestion has been applied

Page 2, Line 6: I suggest adding a period and start a new sentence from "Measurement"
The suggestion has been applied

Page 2, Line 10: "DP" should be "TDP" for Terrestrial Digital Photogrammetry
The correction has been applied

Page 2, Line 12: "rocky outcrops" should be "rock outcrops". Similarly, on Page 7 "rocky slope" and "rocky blocks" should be "rock slope" and "rock block".
The corrections have been applied

Page 2, Line 13: I suggest rephrasing this sentence, something like "A limitation of ground-based remote sensing is related to the survey of complex topography from suboptimal camera or scanner positions, resulting in occlusion zones..."
The correction has been applied

Page 2, line 16: I suggest deleting this sentence. It seems a bit redundant, and not really true, since the next sentences list several examples of the application of RPAS in open-pit mining.
The correction has been applied and the new sentence has been modified as it follows: "There are several photogrammetric studies using RPAS for the geomorphic feature characterization or mapping of the surface extent in open-pit mines (Lamb, 2000; Chen et al., 2015; Shahbazi et al., 2015; Tong et al., 2015; Esposito et al., 2017). Few of them concern the use of RPAS for discontinuity characterization of rock slopes affected by mining activity.....".

Page 2, Line 20: delete "an"
The correction has been applied

Page 2, Line 26: a word is missing "...multicopters results ARE particularly suitable..."
The correction has been applied and the new sentence has been modified as it follows: "In order to analyze rock outcrops, the use of RPAS multicopters results particularly suitable because it allows different geometric configurations for the image acquisition (i.e. zenithal, frontal, oblique).".

Page 2, Line 28: delete "both"
The correction has been applied

Page 2, Line 34: should read " allow only a rough estimation of airborne camera external orientation"
The correction has been applied

Page 3, Line 1: I think the word "accurate" is not appropriate here, because SfM provide accurate models whether they are geo-referenced or not. I suggest rephrasing, something like "in order to geo-reference (or register) 3D models, ..."
The correction has been applied and the new sentence has been modified as it follows: "In order to obtain accurate and georeferenced the 3D models, the use of ground control points (GCPs) surveyed with geodetic GNSS receivers and total station (TS) is generally employed (Francioni et al. 2015).".

Page 3, Line 3: should be "dependent not only ON" (not "from"); same comment at the end of the line
The correction has been applied

Page 3, Line 3: I suggest rephrasing and use "a preliminary rock fall hazard assessment, requested..."
The correction has been applied

Page 3, Line 24: I suggest rephrasing "The bottom of the pit is located at 1,180 meters above sea level (masl) and the top of the excavated rock face is at 1,300 masl.
The correction has been applied and the new sentence has been modified as it follows: "The bottom of the pit is located at 1,180 meters above sea level (m.a.s.l.) and the top of the excavated rock face is at 1,300 m.a.s.l..".

Page 3, Line 29: "compressive tectonic phase WHICH originated..."
The correction has been applied

Page 3, Line 32: "fragile" should read "brittle"?
The correction has been applied

Page 4, Line 1: "motion" should read "displacement" or "offset"?
The correction has been applied and the word "displacement" has been used instead of "motion".

Page 4, Line 3-5: please rephrase with something like "AS involves the oldest LITHOLOGIES of the ..., INCLUDING pre-Alpine..."
The correction has been applied

Page 5, Line 5: would "baseline" be a better terminology for the "two points necessary for the roto-translation of the measured GCPs"?
The correction has not been applied since the terminology "baseline" is correct, especially for GPS measurements, but, in our opinion, less explicative than our long sentence to explain the concept of roto-translation.

Page 7, Line 20: where are the results of block shape and size? Do you mean to say that the results of the kinematic analysis highlight potential for discontinuity-controlled failure mechanism and "therefore the high resolution images and the dense point cloud were analyzed in order to locate possible block source areas"?
We wanted to say that since the traditional kinematic analyses don't allow localization of blocks source areas, the high resolution images were used to localize them. We don't think it is necessary to include the properties of each block, since the paper focus on the 2 bigger and most dangerous blocks. Nevertheless, the sentence wasn't clear, and it has been rewritten it in this way: "The results highlight the potential for blocks to form that may be subject to gravity induced instability but, as previously stated, traditional kinematic analyses do not identify the location of these unstable blocks. Therefore, further analysis of the high resolution images and the dense point cloud was performed in order to locate possible block source areas. More than 20 blocks were deterministically characterized in terms of size, shape and barycentric coordinates, varying from about a cubic meter to a few hundred cubic meters."

Page 7, Line 23: do you mean to say :"In particular, the adopted approach identified two large blocks..."?
The correction has been applied.

Page 7, Line 29-30: I suggest moving this sentence to Line 25, after "high persistence".
The correction has been applied.

Page 8, Line 14: I suggest wording "impossible to measure deterministically"
The correction has been applied.

Page 8, Line 15: I suggest saying that for this reason, persistence is commonly measured as trace length on rock outcrop, and use a more appropriate reference than Einstein et al (1983)
The sentence was rewritten in this way: "The limit of the application of this method is that the discontinuity area is practically impossible to measure deterministically in the field, for this reason persistence is commonly measured as trace length on rock outcrops. Jennings (1970) proposed the following Eq. (2) for persistence calculation starting from trace length values on rock exposure:"

Page 10, Line 10: "slope stability analysis" instead of "slope instability analysis"
The correction has been applied and "instability analysis" has been changed in "stability analysis" in the whole text.

Page 10, Line 25: the reference should be "(Kemeny and Donovan, 2005)"
The correction has been applied.

Figures 3 and 5 captions: "top view" should read "plan view"
The correction has been applied. Figure1 has been similarly modified.

Figure 5 needs to be referenced in the text; the caption should explain that the blue rectangles correspond to the photographs locations; there is not scale nor indication of the north on the figure.
Figure 5 was already referenced in the text (Pag. 5, Line 2). The caption has been modified and the sentence "blue rectangles correspond to the photographs locations, black lines to normals" has been added. A reference scale and the indication of the north have been added to the Figure 5a.

Figure 7: could you please clarify: the caption mention equal area, while the figure shows equal angle. In addition, Figure 7 uses Schmidt method while Figure 8 uses Wulff method.
That was a mistake, the figure has been changed and "Equal angle" corrected
Figure 7 uses Schmidt method since it represents a joint density analysis, while Figure 8 uses Wulff method since it refers to a slope kinematic stability analysis. They are both in theory correct.

Figure 9: "insect photo" should read "inset photo"

The correction has been applied

Figure 13 captions should read "Details of a series of tight discontinuities..."
The correction has been applied
* * *
**Point-by-point response to Reviewer #2**
Dear Authors,
This paper shows not only survey results of complex morphologies using RPAS and SfM-MVS but also a practical application for disaster prevention using those high resolution data, therefore, very interesting. Since detailed measurement procedures, advantages and disadvantages of RPAS and SfM methods are also well explained, I think that this paper is worth to be published. However additional explanations and reconsiderations for the following points should be desired.
Although high resolution 3 dimensional data were obtained using RPAS, does the present stability analysis need that high resolution data? Since the higher resolution of data, the higher costs of data acquisition, processing and handling, appropriate resolution according to the purpose would exist.
We consider high resolution of data always useful in slope stability analyses because it allows the identification and measurement, with high precision and detail, of joints and potential unstable blocks and rock masses at any height above the open pit floor. As written in the Acknowledgments section, that part of the present study has been undertaken within the framework of an agreement with USL1 of Massa and Carrara (Mining Engineering Operative Unit - Department of Prevention) aimed to that purposes. Furthermore, high detail and accurate geometrical data allow deterministic kinematic analyses and the creation of reliable stability models. RPAS photogrammetry is considered a low-cost alternative to traditional remote-sensing techniques given the low cost for digital cameras compared to laser scanners and their ease of use in the field. In this work, for example, we used a light compact Nikon CoolpixA with CMOS sensor that can easily be mounted on a small low-cost RPAS. In this context, the cost of low or high resolution data acquisition are similar, and the decision of decreasing the quality of the data for faster data processing may be adopted in a later stage, depending on the aims of the analyses. In this case for example, a medium-low performance computer (Intel i5 processor with 16 GB RAM) was sufficient for creating the high resolution 3D model. To conclude, costs of data acquisition, processing and handling are not a problem when using RPAS, and this is the reason why they are always more used in engineering geological investigations.
In order to take this aspect of RPAS into considerations we added the following sentences in the text:
Introduction section: "Indeed, RPAS photogrammetry for engineering geological investigations has became widespread mainly because it is a cost-efficient, high flexible and safe technique (Remondino et al., 2011; Siebert and Teizer 2014; Tannant 2015)"
Discussion section: "Nevertheless, in this work the 3D models have been obtained using a medium-low performance computer (Intel i5 CPU @ 3.20 GHz with 16 GB RAM), using images obtained from a light compact digital camera that can easily be mounted on a low cost RPAS. In addition, the use of GCPs overcame the necessity of an expansive IMU system for accurate image alignment. This confirms the reason of the widespread of RPAS for engineering geological investigation, mainly due to its low cost, speed and high safety."

Page 3, lines 10-13: Even though this paper deals with management of natural hazard, detailed description of a real victim would be not necessary in this paper discussing survey method and its application.
The suggestion has been applied and the sentences eliminated.

Figure 4: Although GCPs are located only in the bottom of cliff, is there any effect on the accuracy of 3D model of the cliff?
This is actually a good point. It is true that this problem has repercussion on the final accuracy of the zenithal flight, since the GCPs are located only at the bottom of the cliff. We were aware of that, but the problem was that higher parts of the quarry were inaccessible due to safety reason. In order to overcome this problem GCPs were well spatially distributed, redundant, and the flight altitude was kept low. In addition, it must be considered that a number of photo were convergent and allowed us to build an accurate 3D model even in the surroundings of vertical quarry walls.
On the other hand, it must be considered that additional frontal flights, on the perpendicular to the rock faces, have been executed ad hoc and the related photos used to build a separate frontal model as shown in Figure 6.
In that case we were able to collect GCPs at different heights using a total station, obtaining a very good

model of the cliff. It should be underlined that GCPs for both zenithal and frontal flights are projected in the same reference system. The result of all this is a final root mean square error calculated on the check points (Table 2) of about 6 cm for the zenithal flight and 3 cm for the frontal flights. Therefore, we obtained similar accuracy for the two models, which can be considered adequate for the purpose of the work.

In order to take this problem into consideration, we added the following text in the Discussion section:

"In particular, the final RMSE calculated on the check points (Table 2) was about 6 cm for the zenithal flight and 3 cm for the frontal flight. This small difference is mainly due to the fact that higher parts of the quarry were inaccessible for safety reason, and GCPs of the zenithal flight were only located at the bottom of the cliff. Anyhow, such problem was partially overcame by using GCPs well spatially distributed, redundant and a low flight altitude. In addition, it must be considered that RPAS allow acquisition of a number of convergent photos, that using SfM techniques permit to increase the quality of the model and to build an accurate 3D model even in the surroundings of vertical quarry walls.

Differently, the frontal flights, on the perpendicular to the rock faces, have been executed ad hoc and the related photos used to build a separate frontal model as shown in Fig. 6. In that case the GCPs were collected at different heights using a TS, obtaining a very good model of the cliff. It should be underlined that GCPs for both zenithal and frontal flights are projected in the same reference system. In the end, analysis of the results confirms the good accuracy level of the final model, widely adequate for the purpose of the work."

Figure 6: Although the number of GCPs looks too much, how did you decide their locations and number?

We decided to measure a great number of GCPs (21) because we had to orient, as more accurate as possible, 448 images of a complex morphology. GCPs location has been decided considering a balance between an optimum spatial distribution (Figure 6) both in space, considering the V shape of the "Piastrone" quarry, and elevation from the open pit floor, accessibility (GCPs for the zenithal flight) and easy identification of points on the images (GCPs for the frontal flights).

This aspect of the survey was explained in the Geomatic survey section with the following two sentences:

[revised manuscript text omitted]

---

## Author Response (AR2)

**Author's Response – Review 2 - NHESS-2017-194 Salvini et al. "Use of a remotely piloted aircraft system for hazard assessment in a rocky mining area (Lucca, Italy)"**

**Comment to Reviewers**

**Reviewer 1**
For final publication, the manuscript should be accepted as is.
Considering the Reviewer#1 comment, no modifications to the manuscript have been done.

**Reviewer 2**
**General Comments:**
Many of the reviewers' comments on the first submission have been addressed. When they preferred not to apply the reviewers' comments, the authors generally provided adequate justifications. In my review, I included a few extra minor comments, which the authors should consider mostly to improve clarity of the submission.
At this stage, I recommend minor corrections before publication of the paper. I strongly encourage the authors to proof read the manuscript, in particular Sections 4.2, 4.3, 5 and 6: I made a number of suggestions in the Technical Corrections section below, which could help improve clarity.

**Specific Comments:**
• Section 3.1, first paragraph: is measurement of the exterior orientation of the camera actually used/necessary in the SfM workflow, or was it used as a check or as calibration? I suggest adding a sentence to clarify.
Probably the Reviewer doubt refers to Section 3.2. We have added the following sentence to better clarify the need of the camera exterior orientation: "The exterior orientation of images was necessary to measure the orientation, with respect to the North, and inclination of slopes and discontinuities which are needed for the stability analysis (Firpo et al. 2011)." The cited paper has been added to the reference list.

• Page 11, Line 5: could the authors clarify what they mean by "elaboration speed". Is it the time to generate 3D models, or the time to characterize discontinuities?
It has been better explained that "elaboration speed" refers to the time employed for identifying and characterizing rock discontinuities.

• Page 11, Line 19: "exact geometrical reconstruction" seems a little bit exaggerated, as I do not think that Swedge allows integration of all the details of block geometry.
We agree with the Reviewer suggestion and we have changed the sentence in: "Moreover, the possibility to use the point cloud for obtaining geometrical characteristics of blocks represented a major advantage, because it allowed the correct geometrical reconstruction of a 3D model to be used in specific software for slope stability analysis.".

• It is not always clear which 3D model the authors are referring to. For example, on page 10, Lines 11 to 19 are about both the frontal and the zenithal models. Then, suddenly between Line 10 and 23, the explanation is only about the zenithal model. Between Lines 24 and 26, the sentence is about the frontal model, and then between Lines 26 and 28, the sentence is about both models. I think these paragraphs need to be clarified to make it easier for the reader to follow.
Following the Reviewer suggestion, some paragraphs of Section 5 "Discussion" have been rephrased and other deleted, making the revised form of the text clearer for the reader.

• Similarly, in Section 4.2, which point cloud was used for discontinuity characterization: the frontal model or the zenithal model, or both? Please clarify in the text.
The first paragraph of Section 4.2 has been rephrased. It has been clarified that both the point clouds, derived from frontal and zenithal surveys, were used for discontinuity characterization.

• Figure 8a: why are the authors using the direct toppling method and not the usual planar sliding method? Please modify as appropriate.

That was actually a mistake. We have corrected the images including the correct planar sliding analysis.

• Figure 8b: the analyses are correct. The stereonet show that a number of intersections fall within the failure envelopes, but does not explicitly show which joint set combinations make these intersections. Although the possible intersections are listed in Table 5, it could be useful to plot the great circles of one intersection as an example.
We agree with the suggestion; therefore, we have modified Figure 8b and the relative caption accordingly.

• Section 4.3 and Table 6: I think it is important that the authors provide all the details about the geometry of the block analyzed in Swedge, so that the interested reader can reproduce the results. These parameters could easily be added in Table 6. The following parameters are not provided in the submission: slope dip angle, upper face dip angle and bench width. In addition, it should be made clear which two joints are used for the wedge construction, and if an additional joint is considered as tension crack.
Following the Reviewer suggestion, information about the geometry of the block analysed in Swedge has been added in Fig. 10. Information about Joint 1, Joint 2 and back discontinuity has also been added. The text has been modified for clarifying the joints used for the wedge reconstruction: "In addition, the lateral and rear or back surfaces correspond to geological faults and can be therefore considered fully persistent. In this regard, the western lateral surface has been included in the Swedge model has a discontinuity (Joint 1 in Fig.10; Dip Dir/Dip 307/88), while the back surface has been included as a tension crack (Back discontinuity in Fig.10; Dip Dir/Dip 350/81). It should be also noted that the eastern lateral surface observable in the model (Joint 2 in Fig.10; Dip Dir/Dip 320/74) was also necessary to re-create the block geometry in the software. It was assigned 0° friction angle so as not to induce a resisting force in the simulation."

• Table 1: is the pixel size not supposed to be the sensor size divided by the image size? Please modify if appropriate.
It was a mistake. The correction has been applied. The pixel size is 0.0047 mm.

**Technical Corrections:**
• Page 3, Line 22. Please start a new paragraph at "nevertheless". And, I suggest deleting "nevertheless".
The correction has been applied.

• Page 3, Lines 11 – 22: this paragraph should be the last paragraph of the introduction, as it presents an outline of the paper content.
The correction has been applied.

• Page 7, Line 28: this sentence has not been corrected as indicated in the authors' response; please delete "related to manual placement of GCPs".
The correction has been applied.

• Section 4.2: the 3 first lines are a bit confusing, could the authors please rephrase. I understand that the orientation of 154 discontinuity planes were calculated from the point clouds and plotted on the stereonet, which highlighted four discontinuity sets. And, separately, discontinuity characterization was undertaken in the field based on traditional engineering geological survey.
The 3 first lines have been rephrased as it follows: "A total of 154 discontinuity planes were selected on the 3D point clouds derived from both frontal and zenithal surveys; related attitudes were thus calculated taking into account geographic coordinates of the clouds and then plotted on a stereonet. Analysis of the stereonet allowed identification of four discontinuity sets, whose properties listed in table 3 were obtained in the field through traditional engineering geological survey.".

• Page 8, Line 9: please spell out RMR and GIS as this is the first time they appear in the text. I suggest adding a reference for the RMR.
RMR and GSI were already spelled out and cited in section 3.3.

• Page 8, Line 16: please start a new paragraph at "A discontinuity friction angle"
The correction has been applied.

• Page 8, Line 26: please start a new paragraph at "More than 20"
The correction has been applied.

• Page 10, Line 29: please delete "on the other hand"
The correction has been applied.

• Page 11, Line 21: I suggest starting with a different word, to avoid repetition. For example, the authors could use: "Characterization of the site highlighted a potential significant risk..."
In order to avoid repetition, "In this work" has been eliminated and the sentence rephrased as follows: "Characterization of the site highlighted a potential significant risk for the future workforce due to the presence of two major blocks with potential for sliding.".

• Page 11, Line 22: please delete "In fact"
The correction has been applied.

• Page 11, Line 24: please delete "Moreover"
The correction has been applied.

• Page 11, Line 26: please replace "motion" with "offset"
The correction has been applied.

• Page 11, Line 29: for clarity I suggest replacing "In this stability analysis" by "In the preliminary limit equilibrium analysis...".
The correction has been applied.

• Page 11, Line 31: please delete "the" before "removable"
The correction has been applied.

• Page 11, Line 31: please replace "In particular" with "Indeed"
The correction has been applied.

• Page 11, Line 32: I suggest replacing "The rock bridge failure involves the failure or collapse of the intact rock" with "Rock bridge failure involves fracturing of intact rock..."
The correction has been applied.

• Page 12, Line 3-5: please rephrase and clarify this sentence
The sentence has been rephrased, and the following one has been deleted to avoid repetition. This sentence has been added: "Cohesion induced by the supposed rock bridge leads to an actual stability condition that seems not influenced by the higher dip of the basal plane respect to the friction angle of the surface that, commonly, can causes the failure of a rock block.".

• Page 12, Line 7: I suggest rephrasing this sentence as follows: "The results of this study are consistent with other studies, where failure back-analyses highlighted low percentage of rock bridges (0 and 5%) at the time of failure (e.g., Frayssines....).
The correction has been applied.

• Page 12, Line 10: this last sentence is repetitive, I suggest deleting it.
The correction has been applied.

• Page 12, Lines 11 to 19: for clarity, I suggest rephrasing this paragraph as follows:" The back calculated rock bridge percentage seems in contradiction with the author's field observations and their experience in similar contexts, which suggest that a higher percentage of rock bridges may exist. Hudson and Priest (1983) identified two kinds of persistence relative to impersistent or intermittent joints that should be considered. Differently from impersistent joints, intermittent discontinuities consist of joint segments and intact rock bridges on the same plane. Mauldon (1994) claims that the formation of intermittent joints is geologically unlikely, unless a

preferential direction of weakness exists within the rock mass. In this case, the cohesion of rock bridges along intermittent joints could be much lower than that of the intact rock, and consequently the percentage of rock bridges could be greater. This could be the case of Block A, and the presence of a series of discontinuities with similar dip and dip direction to the basal plane seems to confirm the hypothesis of a preferential plane of weakness due to the geomechanical characteristics of the marble material in that portion of the mining area (Fig. 13)."

The correction has been applied.

• Page 12, Lines 20 to 29: for clarity, I would suggest rephrasing these two paragraphs as follows:" The progressive degradation with time of rock bridge elements could cause a progressive failure mechanism that has the potential to lead to a final rockfall event. This is particularly important in small engineered slopes such as the present one, where the rock mass may be continuously disturbed by excavation activity driving the slope to instability. Such mechanisms of progressive brittle fracturing of rock bridges are not considered in limit equilibrium approaches, where a small content of rock bridges adds significant apparent cohesion to the failure surface (Elmo et al., 2011; 25 Tuckey and Stead, 2016). Therefore, in case of re-opening of mining activities, an in-depth engineering geological analysis together with the installation of a monitoring system for observing the behaviour of the rock mass over time should be considered.

The correction has been applied.

• In the reference list, start a new paragraph before: Priest (1993), and before Sturzenegger and Stead (2009)

The correction has been applied.

• Page 14, Line 30: please delete this keyword, which do not seem to be related to the paper

The correction has been applied.

• Page 17, Line 15: add space between "and" and "modes".

The correction has been applied.

• Table 3: the caption should read "characteristics of the discontinuity sets measured based on traditional manual engineering geological survey in the study area". In addition, please spell out JCS and JRC as this is the first time they appear in the text.

The caption has been modified as suggested by the Reviewer. The JCS and JRC have been spelled out.

[revised manuscript text omitted]

---

## Author Response (AR3)

**Author's Response – Review 3 - NHESS-2017-194 Salvini et al. "Use of a remotely piloted aircraft system for hazard assessment in a rocky mining area (Lucca, Italy)"**

**Comment to Editor**

**Editor**

Thank you for submitting your revised manuscript. I would recommend fixing some minor issues as noted below. Line numbers are those of the change-tracked manuscript in the author responses (nhess-2017-194-author_response-version3).

p. 4, l. 31: Because the term "Tertiary" is not formal anymore, please rephrase this with Paleogene or Neogene.
The sentence has been rephrased as follows: "According to classical interpretation (Carmignani and Kligfield, 1990) AS resulted from a compressive tectonic phase which originated during the continental collision between the Sardinia-Corsica block and the Adria plate (Upper Oligocene-Lower Miocene)."

p. 5, l. 25-29: If the rover data was corrected by post-processing, the term "RTK" is unsuitable. Replace this with "PPK" (post-processing kinematic) or just "with a kinematic mode".
In order to clarify the GNSS RTK procedure, the text has been modified as follows: "The GNSS survey was carried out in real time kinematic (RTK), using geodetic receivers. A reference station was set up, recording continuous signals from the GNSS satellite constellation for more than 3 hours. The positional information obtained by the reference station was then sent to a mobile receiver, using radio modem communication. Each GCP was occupied for at least two minutes with a recording interval equal to 1 second. The reference station coordinates determined using this technique were corrected by post-processing procedures using contemporary data recorded by three permanent GNSS stations (La Spezia, Pieve Fosciana and Pisa) allowing millimetric accuracy. Consequently, the rover data was shifted permitting centimetric accuracy for the eight artificial targets."

p. 8, l. 16: The original term "orientation" would be more suitable than "attitudes"?
The correction has been applied.

p. 8, l. 18: Please capitalize "table 3".
The correction has been applied.

p. 8, l. 26: Please add explanations for "JCond89" and "RQD".
Explanations for JCond89 and RQD has been added: "(GSI=1.5 JCond89 + RQD/2; where JCond89 is the joint condition as defined by Bieniawski, 1989, and RQD is the rock quality designation as defined by Deere, 1963)".

The correction has been applied

[revised manuscript text omitted]